# Size-resolved hygroscopicity and volatility properties of ambient urban aerosol particles measured by the VH-TDMA system in Beijing

Aoyuan Yu [1, 2], Xiaojing Shen [1], Qianli Ma [3], Jiayuan Lu [1], Xinyao Hu [1], Yangmei Zhang [1], Quan Liu [1], Linlin Liang [1], Lei Liu [1], Shuo Liu [1], Hongfei Tong [1], Huizheng Che [1], Xiaoye Zhang [1], Junying Sun [1, *]

[1] State Key Laboratory of Severe Weather & Key Laboratory of Atmospheric Chemistry of CMA, Chinese Academy of Meteorological Sciences, Beijing, 100081, China

[2] University of Chinese Academy of Sciences, Beijing, 100049, China

[3] Lin'an Atmosphere Background National Observation and Research Station, Hangzhou, 311307, China

*Correspondence to*: Junying Sun (jysun@cma.gov.cn)

## Abstract.

Hygroscopicity, volatility and the hygroscopicity of the non-volatile core of submicron ambient aerosol particles with diameters of 50 nm, 80 nm, 110 nm and 150 nm were measured using a Volatility Hygroscopicity Tandem Differential Mobility Analyzer (VH-TDMA) system at a relative humidity of 90 % and a thermal denuder temperature of 270 ℃ from 11 October to 6 November 2023 in Beijing. The mean hygroscopic growth factor (HGF) for particles ranging from 50 nm to 150 nm was $1.15 \pm 0.07$, $1.24 \pm 0.08$, $1.30 \pm 0.09$, and $1.36 \pm 0.10$, respectively, while the mean volatile shrink factor (VSF) was $0.51 \pm 0.05$, $0.55 \pm 0.04$, $0.56 \pm 0.05$, and $0.56 \pm 0.07$, respectively. Both the HGF probability density function (HGF-PDF) and the VSF probability density function (VSF-PDF) for all selected particle sizes exhibited a pronounced bimodal distribution, indicating that the particles were primarily in an external mixing state. Hygroscopicity was observed to increase with particle size in both clean and pollution periods, while volatility decreased slightly with particle size during the clean period, without apparent trend during the pollution period. A positive correlation was identified between hygroscopicity and volatility, as well as between the number fraction of nearly hydrophobic (NH) and non-volatile (NV) particles. Furthermore, this study measured the HGF of the non-volatile core ($HGF_{core}$) of submicron ambient aerosol particles heated at 270 ℃ and derived the HGF of the volatile coating ($HGF_{coating}$). The mean $HGF_{coating}$ for particles ranging from 50 nm to 150 nm particles was $1.17 \pm 0.08$, $1.27 \pm 0.10$, $1.35 \pm 0.10$ and $1.41 \pm 0.10$, respectively, which is 2 % to 7 % higher than the mean HGF for the same particle sizes. The mean $HGF_{core}$ for particles ranging from 50 nm to 150 nm was $1.08 \pm 0.03$, $1.07 \pm 0.03$, $1.07 \pm 0.03$ and $1.09 \pm 0.04$, respectively. The $HGF_{core}$ values were increased when the air mass passed through or originated from the Bohai Sea.

# 1 Introduction

Atmospheric aerosols significantly influence the radiation balance and climate change (IPCC, 2013), play a crucial role in reducing visibility (Zhang et al., 2012), and affect public health (Ou et al., 2020; Lippmann and Albert, 1969). The impact of atmospheric aerosols on global climate, human health, and air quality depends largely on their physicochemical properties (Pöschl, 2005; Buseck and Posfai, 1999). A deep understanding of aerosol particles' physicochemical properties, including hygroscopicity and volatility, is crucial for determining their potential impacts. Hygroscopicity, the ability of aerosols to uptake water vapor (Su et al., 2010), determines the size and optical properties of aerosol particles, thus affecting visibility and direct radiative forcing (Zhao et al., 2019; Xia et al., 2019; Freney et al., 2010). Moreover, hygroscopicity significantly influences CCN activation and cloud droplet formation (Zhang et al., 2023; Tang et al., 2016; Reutter et al., 2009), and the degree of aerosol particles mixing also can be determined by measuring their hygroscopicity distribution (Wang et al., 2019; Ye et al., 2013). The Humidity Tandem Differential Mobility Analyzer (H-TDMA) is extensively used to measure aerosol hygroscopicity in various field studies (Tang et al., 2019; Swietlicki et al., 2008).

Volatility affects the gas-particle distribution of aerosol particles and influences their dry and wet deposition rates (Hakala et al., 2016; Bidleman, 1988), as well as atmospheric lifetime (Huffman et al., 2009). Therefore, studying particle volatility across different regions and environments is essential. The volatility properties of atmospheric aerosols have been investigated using several instruments and techniques (Xu et al., 2019; Jiang et al., 2018; Karnezi et al., 2014). Volatility Tandem Differential Mobility Analyzer (V-TDMA) is one of the commonly instruments used for measuring the volatility of particles. By comparing changes in the diameters of particles before and after heated at a certain temperature, the volatility of particles can be inferred. Most organic compounds evaporate at relatively low temperatures and 98 % of secondary organic aerosols evaporated at approximately 75 °C (An et al., 2007; Tritscher et al., 2011). Ammonium nitrate and ammonium chloride volatilize completely at temperatures below 150 °C, while ammonium sulfate volatilizes entirely between 180-230 °C (Huffman et al., 2008; Feng et al., 2023). However, some compounds like black carbon (BC), soot, sea salt, and crustal materials do not volatilize even at higher temperatures (Massling et al., 2009; Wehner et al., 2009; Zhang et al., 2016). The V-TDMA is usually used to assess the mixing state of typically non-volatile BC particles (Chen et al., 2022; Wehner et al., 2009) and to determine the number, and volume fractions of non-volatile material at a selected temperature for particles (Zhang et al., 2016; Jiang et al., 2018; Ghadikolaei et al., 2020). Studies indicate that BC's internal mixing is higher in winter than in summer (Chen et al., 2022) and increases significantly during the pollution period compared to the cleaner period (Wehner et al., 2009; Wang et al., 2017b).

Although previous studies have focused on the hygroscopicity or volatility of aerosol particles at different sampling locations under varying atmospheric conditions using H-TDMA or V-TDMA (Zhang et al., 2023; Wang et al., 2019; Jiang et al., 2018; Wu et al., 2016; Ye et al., 2013), simultaneous measurements of both properties, particularly in China, have been scarce. The Volatility Hygroscopicity Tandem Differential Mobility Analyzer (VH-TDMA) combines the measurement characteristics of the V-TDMA and H-TDMA systems, providing an

opportunity to simultaneously study the hygroscopicity and volatility of particles, as well as the hygroscopicity of the particle core after the volatilization (Häkkinen et al., 2012; Villani et al., 2008; Johnson et al., 2005). This instrument has been applied in different environments such as oceans (Modini et al., 2009; Johnson et al., 2005), forests (Ristovski et al., 2010; Hong et al., 2014) and urban areas (Enroth et al., 2018; Hakala et al., 2016; Villani et al., 2013) to investigate aerosol mixing states and reveal correlations between hydrophobic and non-volatile

particles (Modini et al., 2009; Ristovski et al., 2010; Enroth et al., 2018). For instance, a study in Budapest (the Republic of Hungary) demonstrated that while hygroscopic particles are generally volatile, hydrophobic and low-volatile particles are predominantly combustion particles emitted from vehicles (Enroth et al., 2018). Zhang et al. (2016) observed strong correlations between hydrophobic and non-volatile particles at a suburban site in Beijing. However, a weak linear relationship was reported between hydrophobic and non-volatile particles in urban Beijing

(Wang et al., 2017a). The four cases in previous study by Villani et al. (2008) showed that gentle thermodesorption has a significant impact on the hygroscopicity of particles at a given RH. It suggests that the loss of volatile material from the particle surface may significantly increase or decrease the hygroscopic growth, depending on the nature of the condensed material. Research on forest aerosol particles shows that hygroscopic materials remain in the particles even after heated at 280 ℃ (Hong et al., 2014). Simultaneous measurement of the hygroscopicity and volatility of

aerosol particles remain scarce. To our knowledge, no study in China has yet focused on the hygroscopicity of particles after heating. Investigating the hygroscopicity of the particle core after the volatilization is crucial for understanding their composition and sources. In recent years, China has gained significant progress in air pollution control. However, atmospheric aerosol pollution events still frequently occur in major cities (Zhang et al., 2018; Wang et al., 2019; Fan et al., 2020a; Miao et al., 2021; Lu et al., 2024). Beijing, the capital of China, regularly

experiences haze events due to its large population, substantial industrial emissions, and an increasing number of vehicles (Zhong et al., 2018; Huang et al., 2021; Yang et al., 2023). In this study, we utilized the VH-TDMA system for the first time to investigate the hygroscopicity and volatility, and the hygroscopicity of the non-volatile core of submicron aerosols in urban Beijing during the autumn of 2023. We explored the variations in hygroscopicity, volatility, and particle mixing states under different atmospheric conditions and further analyzed the relationship

between the hygroscopicity and volatility of aerosol particles. We also studied the hygroscopicity of non-volatility particles after heated at 270 °C, a temperature at which atmospheric sulfates, nitrates, and most organic compounds are typically volatile, whereas soot, sea salt or certain organic polymers would remain refractory (Enroth et al., 2018; Johnson et al., 2005).

## 2 Sampling site and experimental setup

### 2.1 Sampling site

This study was conducted in the Beijing urban area, with the sampling site located at the Chinese Academy of Meteorological Sciences (CAMS; 39.97°N, 116.37°E), situated on the campus of the China Meteorological Administration in Beijing, China. The site is located between the 2nd and 3rd ring roads of Beijing. To the west of the site is a major road with heavy traffic, located less than 200 m away, and the site is predominantly affected by residential and traffic emissions (Wang et al., 2018). Measurements of aerosol hygroscopicity and volatility were conducted from 11 October to 6 November 2023 in a laboratory on the second floor (~ 6 m above the ground) of the CAMS building. The particle number size distribution and mass concentrations of the main chemical composition of non-refractory $PM_1$ were simultaneously measured on the observation platform located on the roof of the CAMS building, where ambient aerosols were pumped through a $PM_{10}$ impactor (URG Corporation at a flow rate of 16.7 L/min) and were then dried to less than 30 % RH using an automatic aerosol dryer. The dried sample was then split through a manifold to different instruments, including the Tandem Scanning Mobility Particle Sizer (TSMPS, TROPOS, Germany) and a High Resolution Time-of-Flight Aerosol Mass Spectrometer (HR-ToF-AMS, Aerodyne Research, Inc., USA). For the VH-TDMA system, a silica dryer and a Nafion dryer in series were used to dry the sample air to less than 30 % RH. A more detailed description of the instruments used in this study is provided below.

### 2.2 Experimental setup

The schematic diagram of the Volatility Hygroscopicity Tandem Differential Mobility Analyzer (VH-TDMA) (TROPOS, Germany) is shown in Fig. 1. The VH-TDMA components mainly consist of three medium-Hauke type differential mobility analyzers (DMA1, DMA2, and DMA3, TROPOS, Germany), two CPCs (CPC1 and CPC2; CPC 3772, TSI, USA), an X-ray aerosol neutralizer (Model 3088; TSI Inc., USA), custom-made Nafion dryers, thermal denuders (TD), and humidifiers. DMA1 is utilized for the size selection of monodisperse particles, while DMA2+CPC1, DMA3+CPC2 are used to measure thermally treated and humidified aerosols, respectively. The thermal denuder includes four columns, whose temperature can be individually controlled from ambient temperature up to 350 °C, whereas the RH of the aerosol and sheath air could be controlled up to 90 %, which is monitored by

two humidity and temperature probe (HMP7, Vaisala, Finland) and regulated by adjusting the proportion of dry and wet air using mass flow controllers (MKS, USA). DMA3 and the humidifiers are placed inside a temperature-controlled box to maintain a temperature stable. The system also includes a nebulizer setup for routine hygroscopicity check using ammonium sulfate solution, as well as ball valves, which control the system operation modes through a home-made program.

In this study, VH-TDMA was alternately operated in the combinations of V-TDMA mode and H-TDMA mode or V-TDMA mode and VH-TDMA mode. In the first combination, the monodisperse particles from DMA1 were equally split to two flows. One flow went to the thermal denuder set at 270 ℃, then to the DMA2+CPC1 for measuring the size distribution of heated particles. The other flow went to the aerosol humidifier set at 90 % RH, then to the DMA3+CPC2 for measuring the size distribution of humidified particles. Therefore, the volatility and hygroscopicity of particles were simultaneously measured in this combination. Similarly, in the latter combination, the particles from DMA1 went through the thermal denuder first, then equally split to two flows. One flow directly went to the DMA2+CPC1 for measuring the size distribution of heated particles, and another flow went to the aerosol humidifier at 90 % RH, then to the DMA3+CPC2 for measuring hygroscopicity of these particles after heating. The sample air for the VH-TDMA was dried to an RH of less than 30% using a silica dryer and a Nafion dryer in series and was then neutralized with an X-ray aerosol neutralizer. All DMAs were operated with a sheath-to-aerosol flow ratio of 5:1. For DMA1, the sheath flow was 10 L min$^{-1}$ with an aerosol flow of 2 L min$^{-1}$, while for DMA2 and DMA3, the sheath flow was 5 L min$^{-1}$ and the aerosol flow was 1 L min$^{-1}$. The particle diameter selected by DMA1 were 50 nm, 80 nm, 110 nm, and 150 nm, which were scanned using different size ranges after hydration or volatilization. Detailed information was provided in Table S1 in the supplement. Each particle size was sampled for approximately 15 minutes, including about 8 minutes for measurement and 6 minutes for system flushing between combination change or size change, with each cycle lasting about 1 h. The thermal denuder is well-insulated cylindrical metal tube, maintained a consistent heating temperature at $270 \pm 0.5$ ℃. The RH of the aerosol flow and sheath flow in the humidification system was set to 90 %, and maintained at $90 \pm 1$ %. The hygroscopicity of 100 nm ammonium sulphate ($(NH_4)_2SO_4$) particles generated by nebulizer were measured regularly to check the accuracy of RH measurement or system performance. Size calibration was also conducted for DMAs using the Polystyrene Latex Spheres (PSL) particles with a nominal size of 203 nm. Data inversion was accomplished using the TDMAinv program package (Gysel et al., 2009) in IGOR Pro software.

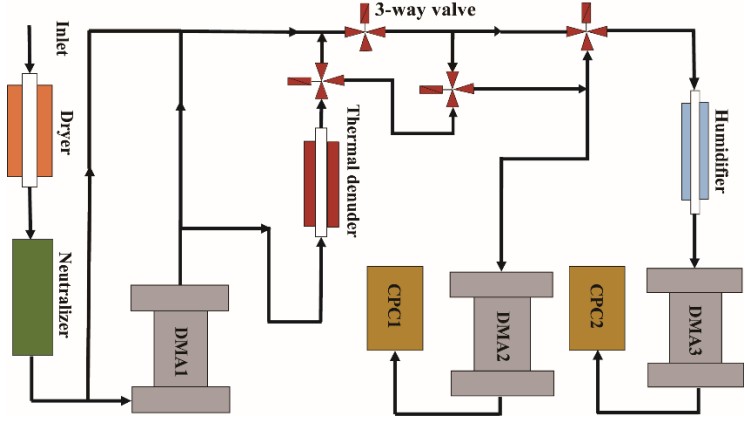

**Figure 1.** The schematic diagram of the VH-TDMA.

The particle number size distribution (PNSD) in the range of 3-850 nm is measured by a twin scanning mobility particle sizer (TSMPS, TROPOS, Germany). This system is comprised by two differential mobility analyzers (DMA and UDMA, TROPOS, Germany) and two CPCs (CPC 3772 and ultrafine CPC 3776, TSI Inc., St Paul, USA). The combination of CPCs and DMAs can measure the PNSD in the size ranges of 3-80 nm and 40-850 nm, respectively. The combined PNSD covers the size range of 3-850 nm with 40 size bins in a 10-minute time resolution. More information about the instrument setup and data evaluation can be referred to Shen et al. (2018).

In addition, a high-resolution time-of-flight aerosol mass spectrometer (HR-ToF-AMS, Aerodyne Research, Inc., USA) was used to measure the mass concentrations of the main chemical composition of non-refractory $PM_1$, including organic matter, sulfate, nitrate, ammonium and chloride from 24 October to 6 November 2023, with a time resolution of 5 min. For more detailed information on the working principles of the instrument and data processing, refer to Zhang et al. (2012).

The mass concentrations of hourly $PM_{2.5}$ and $PM_{10}$ in this study were from GuanYuan Air Quality Monitoring Site, China National Environmental Monitoring Center (CNEMC, http://www. cnemc.cn), located ~3 km from the CAMS site. Hourly meteorological data used in this study were from the Haidian National Basic Meteorological Station (station No.54399) of Beijing Meteorological Bureau, located ~ 5 km northwest of the CAMS site. The GDAS data from the National Center for Environmental Prediction (NCEP) were used to derive 48 h backward trajectory at an altitude of 500 m above ground level of the CAMS station, using the trajectory calculation software Trajstat developed by Wang et al. ( 2011) at the Chinese Academy of Meteorological Sciences. All data in this study were in Beijing Time (UTC + 8).

# 3 Data analysis

## 3.1 Parameters derived from VH-TDMA

The hygroscopic growth factor (HGF) is defined as the ratio of the particle's electrical mobility diameter under a given relative humidity to its diameter under dry condition at room temperature:

$$HGF = \frac{D_p(T_{room}, RH)}{D_p(T_{room}, RH_{dry})} \tag{1}.$$

In this study, $D_p(T_{room}, RH_{dry})$ is the particle diameter selected by DMA1 at room temperature and an RH below 30 %, while $D_p(T_{room}, RH)$ is the diameter of the same particle after being humidified at an RH of 90 %.

The volatile shrink factor (VSF) is defined as the ratio of the particle's electrical mobility diameter under a given temperature to its diameter at room temperature under dry condition:

$$VSF = \frac{D_p(T, RH_{dry})}{D_p(T_{room}, RH_{dry})} \tag{2}.$$

In this study, $D_p(T, RH_{dry})$ is the particle diameter at a set heating temperature of 270 °C, which lies in the middle of the optimum temperature range of 250-300 °C to avoid size dependent effect (Villani et al., 2007).

Volatility hygroscopic growth factor (VHGF) is defined as the ratio of a particle's electrical mobility diameter after humidification following heating to its diameter under dry condition:

$$VHGF = \frac{D_p(T, RH)}{D_p(T_{room}, RH_{dry})} \tag{3}.$$

In this study, $D_p(T, RH)$ is the diameter of particle after humidification at 90% RH following heating at 270 °C.

The hygroscopic growth factor of the non-volatile core of aerosol particle after heating, denoted as $HGF_{core}$ are derived as (Hong et al., 2014):

$$HGF_{core} = \frac{D_p(T, RH)}{D_p(T, RH_{dry})} = \frac{VHGF}{VSF} \tag{4}.$$

The volume fraction remaining (VFR) after heating was calculated from the VSF assuming spherical shape of particles both before and after heating:

$$VFR = \frac{D_P^3(T, RH_{dry})}{D_P^3(T_{room}, RH_{dry})} = VSF^3 \tag{5}.$$

Thus, 1-VFR is the volume fraction of the volatile coating for particles.

According to the Zdanovskii - Stokes - Robinson (ZSR) relation (Stokes and Robinson, 1966), the hygroscopic growth factor of a particle can be estimated from the HGFs of its components according to their volume fractions (Gysel et al., 2009; Swietlicki et al., 2008). Therefore, the mean hygroscopic growth factor of volatile fraction of particles in this study can be calculated as follows:

$$HGF_{mean}^3 = VFR * HGF_{core}^3 + (1 - VFR) * HGF_{coating}^3 \tag{6},$$

where $HGF_{mean}$, $HGF_{core}$, and $HGF_{coating}$ are the mean hygroscopic growth factors at 90 % RH for dry particles, the non-volatile core, and the volatile coating, respectively.

The HGF-PDF, VSF-PDF and VHGF-PDF (volatile hygroscopic growth factor probability density function) were retrieved from the measured number size distributions of humidified aerosol particles, volatile aerosol particles, and particles following humidification after heating. $c$ (HGF, $D_p$), $c$ (VSF, $D_p$), and $c$ (VHGF, $D_p$) are the probability density functions of the particle HGF, VSF and VHGF, respectively. They describe the probability that particles with a dry diameter $D_p$ exhibit a certain HGF, VSF, or VHGF under a defined treatment. The inverted probability density functions are normalized such that the total probability equals unity: $\int_0^\infty c(GF, D_p) dGF = 1$ (Gysel et al., 2009). The number-weighted mean hygroscopic growth factor was calculated as follows:

$$HGF_{mean} = \int_0^\infty HGF\, c\left(HGF, D_p\right) dHGF \tag{7}.$$

The mean VSF ($VSF_{mean}$) and mean VHGF ($VHGF_{mean}$) were calculated same as the $HGF_{mean}$.

According to the κ-Köhler theory (Petters and Kreidenweis, 2007), the single hygroscopic parameter $\kappa$ can be calculated as follow:

$$\kappa = (HGF^3 - 1)[\frac{1}{RH} \exp\left(\frac{4\sigma M_w}{RT\rho_w D_p\, HGF}\right) - 1] \tag{8},$$

where RH is the relative humidity, HGF is the mean hygroscopic growth factor from Eq. (7), $D_p$ is the diameter of dry particles, σ is the droplet surface tension (assumed to be that of pure water, σ = 0.0728 N m$^{-2}$), $M_w$ is the molecular mass of water, $R$ is the universal gas constant, T and $\rho_w$ are the temperature (in K) and density of water, respectively.

Based on the measured HGF-PDFs and VSF-PDFs, aerosol particles were classified into two hygroscopic groups and two volatile groups in this study, i.e. nearly hydrophobic (NH) mode particles (HGF ≤ 1.2) and more hygroscopic (MH) mode particles (HGF > 1.2), as well as non-volatile (NV) mode particles (VSF ≥ 0.8) and very volatile (VV) mode particles (VSF < 0.8).

The number fraction (NF) is defined as the proportion of the number of particles with HGF in the boundary of (a, b) and computed as follow:

$$NF = \int_a^b c(HGF, D_p) dHGF \tag{9}.$$

The number of particles with VSF in the boundary was calculated same as the number of particles with HGF.

The standard deviation (σ) of the HGF-PDF is used as a measure for the spread of hygroscopic growth factors (Liu et al., 2011; Sjogren et al., 2008), is defined as follow:

$$\sigma_{HGF-PDF} = [\int_0^\infty (HGF - HGF_{mean})^2 c(HGF, D_p) dHGF]^{1/2} \tag{10}.$$

$\sigma_{HGF-PDF} \leq 0.10$ indicates that particles are in a state of internal mixing with limited spread of hygroscopic growth factors, while a $\sigma_{HGF-PDF} \geq 0.15$ indicates substantial external mixing (Sjogren et al., 2008).

## 3.2 Interference of multi-charged aerosol particles

The DMA selects aerosol particles based on their electrical mobility, which depends on both their physical diameter and the charge they carry. Therefore, aerosol particles selected by DMA1 can have different physical diameters if they carry different numbers of charges, which could potential lead to inaccuracies in the measured hygroscopic growth factor (Gysel et al., 2009; Duplissy et al., 2009). Based on the simultaneously measured particle number size distribution of the dry ambient aerosol, the charge probability and the transfer function of DMA, the

number fraction of particles with different charges was calculated using Petter's program and shown in Fig. S1 (Petters, et al., 2018; Wiedensholer, 1988). The results indicate that the number fraction of singly charged particles among the 50 nm, 80 nm, 110 nm, and 150 nm particles selected by DMA1 was 94 %, 89 %, 84 %, and 84 %, respectively, during the clean period, whereas during the pollution periods, the value were 91 %, 81 %, 74 %, and 72 %, respectively. Duplissy et al. (2009) mentioned that when the ratio of singly charged aerosol particles exceeds

80% of the total particles, HTDMA data analysis is straightforward and not biased by the multiply charged aerosol particles. Since singly charged particles dominated in our study, the effect of multiply charged particles should be insignificant, although the number fraction of singly charged particles for some sizes during pollution periods was slightly lower than 80 %.

       To further investigate the impact of multi-charged particles on the hygroscopicity of singly charged particles, the

apparent electrical mobility diameters of doubly and triply charged particles after hygroscopic growth were calculated and shown in Table S2 and Fig. S2 in the supplement. During the calculation, we considered the variation in hygroscopicity with both particle diameter and pollution level. Suppose that multi-charged particles have the same HGF as singly charged particles with the same electrical mobility, the apparent mobility diameter of doubly or triply charged particle was a few percent smaller (less than 5% for the size range of 50-150 nm) than that of

singly charged particle after hydration. This difference was increased for particles with higher charges and larger particle size (Table S2). However, suppose HGF of the multi-charged particles equals that of particles with the same physical diameter under different pollution level, the apparent mobility diameter of doubly or triply charged particle could be either larger or smaller by a few percent (less than 12 % for size range of 50-150 nm) compared to singly charged particle after hydration. In this case, the difference was larger for smaller particle sizes (Table S2, Fig. S2),

which is contrary to the results obtained under the above assumption. It is worth noting that the HGF of multi-charged particles was interpolated from the measured size-resolved HGF. When a particle's physical diameter

exceeded the maximum of the measured size (150 nm), HGF (150 nm) was used in the latter assumption. For particles with the same electrical mobility but different charges, it is clear that the apparent mobility diameter of doubly or triply charged particles after hydration could be either larger or smaller than that of singly charged particles after hydration, which is related to the mobility size and the assumed hygroscopicity of multi-charged particles.

## 4 Results and discussion

### 4.1. Meteorological conditions during the sampling period

Time series of hourly meteorological parameters and $PM_{2.5}$, $PM_{10}$ mass concentrations from 11 October to 6 November 2023 were shown in Fig. 2a. The average mass concentrations of $PM_{2.5}$ and $PM_{10}$ were $50.7 \pm 51.7$ μg $m^{-3}$ and $95.0 \pm 72.2$ μg $m^{-3}$, respectively. Temperature and relative humidity ranged from 1.8 to 28.4 °C and 14 to 100 %, with average of $13.9 \pm 5.1$ °C and $66.3 \pm 25.8$ %, respectively. The average wind speed was $0.9 \pm 0.9$ m $s^{-1}$ during the sampling period, predominantly from the northeast. Pollution periods were defined as time when hourly $PM_{2.5}$ concentrations exceeded 75 μg $m^{-3}$, based on Grade II standard values of Chinese Ambient Air Quality standards (GB3095-2012). Three pollution periods are identified as: $P_I$ (12 October, 9:00–13 October, 9:00), $P_{II}$ (23 October, 9:00–25 October, 13:00), and $P_{III}$ (28 October, 10:00–2 November, 17:00). It is worth to mention that more than 90 % time period of an orange alert for severe air pollution (from 30 October, 12:00 to 2 November, 24:00) was included in $P_{III}$. We classified the periods with hourly $PM_{2.5}$ concentrations below 30 μg $m^{-3}$ during new particle formation (NPF) event days as clean periods (13 October, 11:00 to16 October, 23:00; 18 October, 05:00 to 21 October, 23:00; 26 October, 00:00 to 23:00; 03 November, 00:00 to 23:00; and 06 November, 00:00 to 23:00), during which eleven NPF events were observed (Fig. 2b). The mean PNSD distribution during the NPF days and PNSD during each NPF day were shown in Fig. S6, which indicates that NPF typically occurred around 10:00.

During the pollution period, average concentrations of $PM_{10}$ and $PM_{2.5}$ were $185.2 \pm 41.6$ μg $m^{-3}$ and $118.7 \pm 33.1$ μg $m^{-3}$, respectively, compared to $32.3 \pm 24.3$ μg $m^{-3}$ for $PM_{10}$ and $9.2 \pm 6.5$ μg $m^{-3}$ for $PM_{2.5}$ during the clean period. The average wind speed during the pollution period ($0.7 \pm 0.6$ m $s^{-1}$) was lower than that during the clean period ($1.2 \pm 1.1$m $s^{-1}$), which facilitates the accumulation of pollutants (Wehner et al., 2008). Analysis of air mass back trajectories during the pollution period (Fig. S3b) reveals that 76 % of the air masses passed through densely populated southern region of Beijing (such as Hebei, northwestern Shandong and northern Shanxi) and 98.5 % of the air masses traveled below 1.0 km before reaching the sampling site. In contrast, air masses originated from the northwest during the clean period (Fig. S3a). Air masses from the northwest moved more quickly and accumulated fewer pollutants during the transport before reaching the sampling site.

## 4.2 Time series of HGF- PDF、VSF- PDF and averages of hygroscopicity and volatility

Figure 2 and S4 show the time series of HGF-PDFs and VSF-PDFs for different size particles throughout the sampling period. Both HGF-PDFs and VSF-PDFs exhibited clearly bimodal distributions for 50 nm and 150 nm particles (Fig. 2), similar for 80 nm and 110 nm particles shown in Fig. S4. As mentioned above, the lower (HGF ≤ 1.2) and higher modes (HGF > 1.2) of HGF-PDFs are referred to as NH and MH modes, respectively. The lower (VSF < 0.8) and higher mode (VSF ≥ 0.8) of VSF-PDFs correspond to the VV and NV mode, respectively. As shown in Fig. 2c-f, the MH mode for HGF-PDFs and the VV mode for VSF-PDFs showed larger variations than the NH mode and NV mode during the course of the sampling period. However, HGF-PDFs and VSF-PDFs for 150 nm particles exhibited smaller variations and higher proportions of more hygroscopic (MH) and very volatile (VV) modes during the pollution period, especially during $P_{III}$ (Fig. 2d and 2f). The variations in the HGF-PDFs and VSF-PDFs over time could be influenced by different emission sources, atmospheric processes, meteorological factors and air mass origins (Enroth et al., 2018). Although the unimodal or bimodal distribution of the HGF-PDFs and VSF-PDFs could give some information on the mixing state of particles, the standard deviation of the HGF-PDF is used to quantify the degree of external mixing of particles (Sjogren et al., 2008; Liu et al., 2011). A value of $\sigma_{HGF\text{-}PDF} \geq 0.15$ indicating an externally mixed state, as defined by Sjogren et al. (2008). In this study, more than 85 % of HGF-PDF for 80 nm, 110 nm and 150 nm aerosol particles was exhibited a $\sigma_{HGF\text{-}PDF} \geqslant 0.15$, while 46 % of the 50 nm aerosol particles. The mean $\sigma_{HGF\text{-}PDF}$ for 50 nm, 80 nm, 110 nm, and 150 nm particles was $0.15 \pm 0.03$, $0.19 \pm 0.04$, $0.20 \pm 0.04$, and $0.20 \pm 0.04$, respectively. These results indicate a stronger degree of external mixing for larger particles.

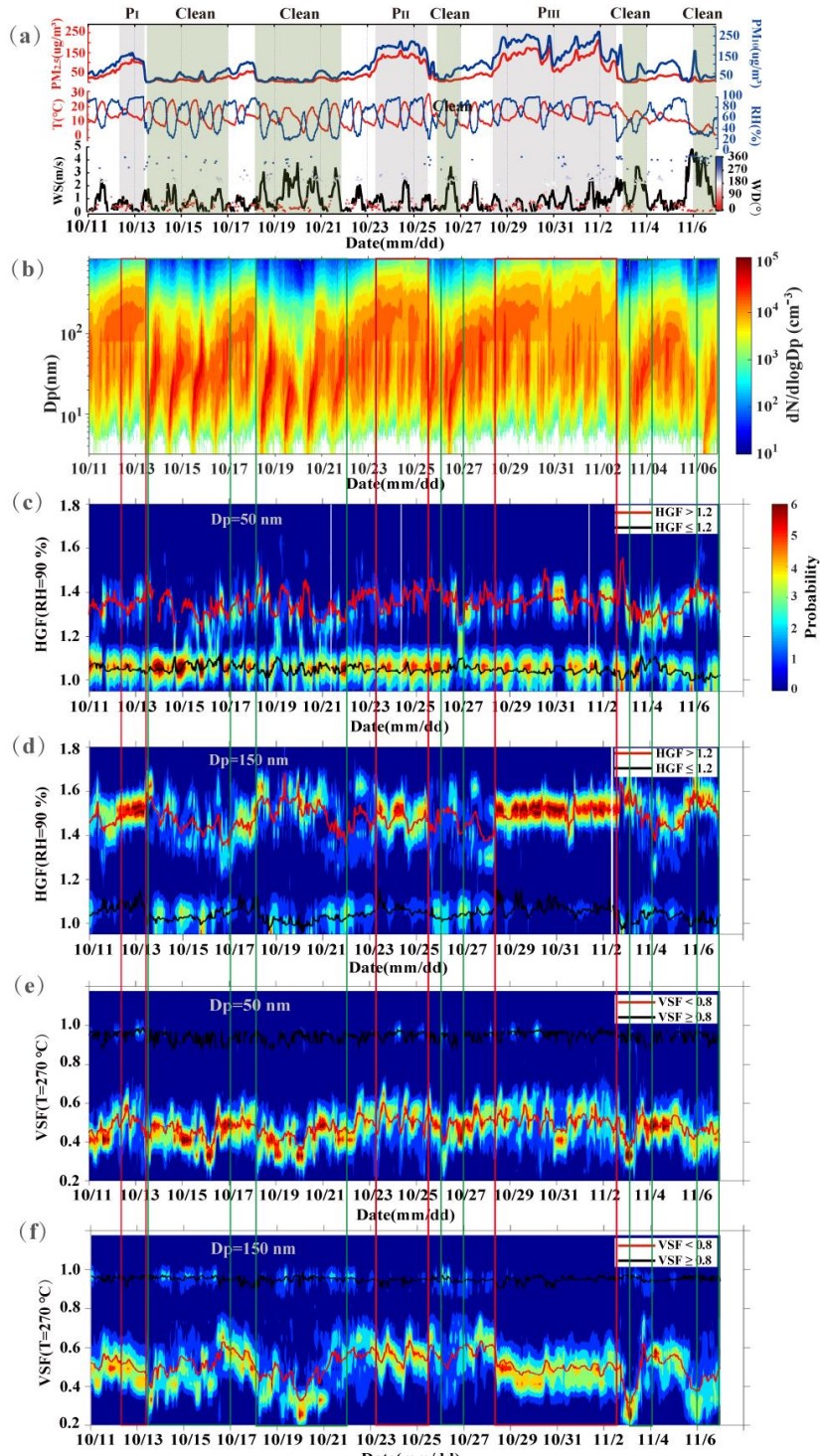

**Figure 2.** Time series of PM$_{2.5}$ and PM$_{10}$ mass concentrations and meteorological parameters (a), PNSD from TSMPS (b), hygroscopic growth factor probability density function (HGF-PDF) for 50 nm (c) and 150 nm (d) particles at RH = 90 %, and volatile shrink factor probability density function (VSF-PDF) for 50 nm (e) and 150 nm (f) particles at $T$ = 270 °C. In the HGF-PDFs, the blackline represents the mean HGF of the nearly hydrophobic mode (HGF$_{NH}$), and the red line represents the mean HGF of the more hygroscopic mode (HGF$_{MH}$). In the VSF-PDFs, the blackline represents the mean VSF of the non-volatility mode (VSF$_{NV}$), and the red line represents the mean VSF of the very volatility mode (VSF$_{VV}$).

Figure 3 displays the size-resolved mean HGF-PDFs and VSF-PDFs for the sampling periods. As mentioned above, both the HGF-PDFs and VSF-PDFs exhibited a clearly bimodal distribution. Table 1 summarizes the size-resolved mean HGF, VSF, $\kappa$, the number fraction of nearly hydrophobic particles, the number fraction of non-volatile particles, and the growth spread factor ($\sigma_{HGF-PDF}$) of HGF-PDF during different periods. The averaged HGF-PDFs at 90 % RH typically show a minimum at HGF around 1.2 (Fig. 3a), which is why we use HGF = 1.2 as the separation between NH and MH mode in this study. The contribution of the NH mode in the HGF-PDF decreased with increasing particle diameter, with the average number fraction of NH ($NF_{NH}$) decreasing monotonically from 66 % to 28 % as particle diameter increased from 50 nm to 150 nm (Fig. 3a and Table 1). The peak values of the NH mode remained around 1.06, independent of particle diameter. The peak values of the MH mode increase from 1.33 to 1.52 as particle size increased from 50 nm to 150 nm. Previous study has found that the proportion of the NH mode is higher in Beijing than that in Shanghai and Guangzhou, likely due to higher number of vehicles, resulting from higher emissions of hydrophobic primary aerosols in Beijing (Ye et al., 2013). The strength of the MH mode increased with particle size, indicating that the proportion of strongly hygroscopic particles gradually increases with particle size. MH-mode particles are mainly from secondary formation or the aging of primary particles (Wang et al., 2019).

The averaged VSF-PDFs typically showed a minimum at VSF around 0.8 (Fig. 3b), thus we use VSF = 0.8 as the separation between NV and VV mode. The contribution of the NV mode in the VSF-PDF increased with particle diameter, and the average number fraction of NV ($NF_{NV}$) increased from 8 % to 14 % (Fig. 3b). This trend aligns with observations in Beijing and Shanghai (Jiang et al., 2018; Wehner et al., 2009). Particles in the NV mode contained non-volatile species, likely BC, highly oxidized organics or polymers (Hong et al., 2014). Larger particles showed stronger NV modes, which are significantly influenced by highly oxidized organics aerosols, and volatility is generally inversely correlated with the oxidation state (Wang et al., 2017a). The size-resolved VSF-PDFs exhibited an NV mode peak near 0.95 and a broad, approximately symmetric VV mode peak ranging between 0.47 and 0.52 for all measured particle diameters. Unlike the MH mode, the peak values in the VV mode showed no clear dependency on diameter. Additionally, $NF_{NV}$ was observed to be lower than $NF_{NH}$ in this study, indicating a significant presence of hydrophobic but volatile particles. Previous researches suggest that $NF_{NH}$ and $NF_{NV}$ are predominantly influenced by the same component, associated with newly emitted combustion particles (Liu et al., 2013; Cheung et al., 2016; Zhang et al., 2016). The lower proportion of $NF_{NV}$ observed in this study could be attributed to Beijing's recent restrictions on coal burning, which have led to a reduction in non-volatile and non-hygroscopic carbon emissions (Hu et al., 2024). Currently, traffic emissions are the primary pollution source in

Beijing, characterized by freshly emitted organics from vehicles with a low oxidation state, high volatility, and low

hygroscopicity (Tiitta et al., 2010; Feng et al., 2023; Liu et al., 2023).

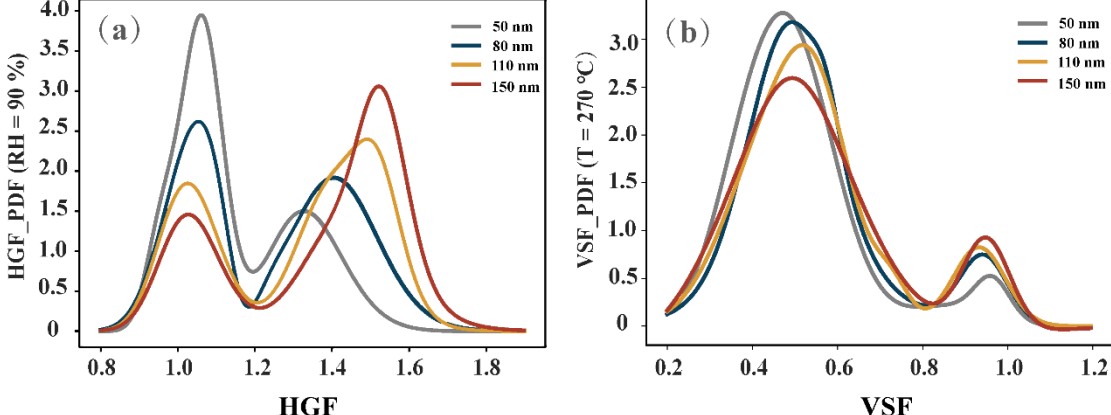

**Figure 3.** Size-resolved mean hygroscopicity growth factor probability density functions (HGF-PDFs) (a) and volatility shrink factor probability density function (VSF-PDFs) (b) for particles ranging from 50 nm to150 nm throughout the sampling period.

**Table 1.** Summary of the hygroscopicity and volatility parameters throughout the sampling period from 11 October to 6 November 2023, as well as during the clean and pollution periods.

| | Particles property | 50 nm Mean ± SD | 80 nm Mean ± SD | 110 nm Mean ± SD | 150 nm Mean ± SD |
|---|---|---|---|---|---|
| **Total** | $NF_{NV}$ | 0.08 ± 0.06 | 0.11 ± 0.06 | 0.13 ± 0.06 | 0.14 ± 0.07 |
| | VSF | 0.51 ± 0.05 | 0.55 ± 0.04 | 0.56 ± 0.05 | 0.56 ± 0.07 |
| | $NF_{NH}$ | 0.66 ± 0.20 | 0.47 ± 0.20 | 0.35 ± 0.18 | 0.28 ± 0.17 |
| | HGF | 1.15 ± 0.07 | 1.24 ± 0.08 | 1.30 ± 0.09 | 1.36 ± 0.10 |
| | κ | 0.062 ± 0.032 | 0.103 ± 0.043 | 0.138 ± 0.050 | 0.174 ± 0.058 |
| | $\sigma_{HGF\text{-}PDF}$ | 0.15 ± 0.03 | 0.19 ± 0.04 | 0.20 ± 0.04 | 0.20 ± 0.04 |
| **Clean** | $NF_{NV}$ | 0.05 ± 0.04 | 0.10 ± 0.06 | 0.12 ± 0.06 | 0.15 ± 0.07 |
| | VSF | 0.49 ± 0.05 | 0.55 ± 0.05 | 0.57 ± 0.07 | 0.57 ± 0.08 |
| | $NF_{NH}$ | 0.65 ± 0.20 | 0.55 ± 0.20 | 0.45 ± 0.19 | 0.38 ± 0.18 |
| | HGF | 1.15 ± 0.07 | 1.21 ± 0.09 | 1.26 ± 0.10 | 1.32 ± 0.11 |
| | κ | 0.061 ± 0.032 | 0.088 ± 0.047 | 0.118 ± 0.056 | 0.148 ± 0.062 |
| | $\sigma_{HGF\text{-}PDF}$ | 0.14 ± 0.04 | 0.19 ± 0.04 | 0.22 ± 0.04 | 0.23 ± 0.04 |
| **Pollution** | $NF_{NV}$ | 0.10 ± 0.07 | 0.12 ± 0.05 | 0.11 ± 0.04 | 0.09 ± 0.03 |
| | VSF | 0.56 ± 0.05 | 0.58 ± 0.04 | 0.57 ± 0.05 | 0.55 ± 0.05 |
| | $NF_{NH}$ | 0.67 ± 0.21 | 0.40 ± 0.16 | 0.23 ± 0.10 | 0.14 ± 0.06 |
| | HGF | 1.15 ± 0.07 | 1.28 ± 0.07 | 1.36 ± 0.05 | 1.44 ± 0.04 |
| | κ | 0.060 ± 0.033 | 0.121 ± 0.037 | 0.172 ± 0.032 | 0.218 ± 0.029 |
| | $\sigma_{HGF\text{-}PDF}$ | 0.15 ± 0.03 | 0.19 ± 0.02 | 0.18 ± 0.03 | 0.16 ± 0.03 |

## 4.3 Diurnal variations of single hygroscopicity parameter and VSF$_{mean}$

Figure 4a and 4b display the diurnal variation of the mean single hygroscopicity parameter ($\kappa_{mean}$) for particles of different sizes during the clean and the pollution periods, respectively. The $\kappa_{mean}$ shows similar diurnal patterns during both periods, increasing in the afternoon and early morning, and decreasing at noon and during the evening rush hours. However, peak values of $\kappa_{mean}$ occurred at different times: in the late afternoon (15:00-16:00) during the clean period (Fig. 4a) and in the early morning (05:00-06:00) during the pollution period (Fig. 4b). This discrepancy is likely due to an increase in aged particles from daytime photochemical reactions during the clean period, consistent with findings from the same sampling site (Zhang et al., 2023). During the pollution period, the early morning peak in $\kappa_{mean}$ can be attributed to more active heterogeneous hydrolysis reactions of $N_2O_5$, which rapidly convert to hygroscopic nitrates at lower temperatures (Zhou et al., 2018; Zhao et al., 2017), coupled with lower wind speeds and higher humidity that enhance condensation on aerosol particles. This results in the transformation of low-hygroscopicity primary particles into more hygroscopic and internally mixed aerosol particles (Fan et al., 2020b; Zhang et al., 2016). The two low $\kappa_{mean}$ values occurred at noon (12:00-13:00) and evening rush time (19:00-20:00) (Fig. 4). During these periods, the average organic mass fraction of PM$_1$ increased (Fig. S5). The decrease in hygroscopicity around noon was more pronounced for Aitken mode particles compared to accumulation mode particles, which could be attributed to the cooking emissions. HGF values of 1.13 and 1.17 were observed for 50 nm and 80 nm particles respectively at noon during the clean period, while HGF values of 1.10 and 1.25 were observed during the pollution period. These findings are consistent with the previous results that reported HGF value of $1.17 \pm 0.11$ for 40 nm and 80 nm particles during cooking activities in Beijing (Ren et al., 2023). The lowest particle hygroscopicity was observed at evening rush time, likely due to the reduction in the boundary layer at night, which traps local pollutants from cooking and vehicle emissions, leading to decreases in aerosol hygroscopicity (Zhang et al., 2016).

Figures 4c and 4d show notable differences in the diurnal variations of $\kappa$-PDF for 50 nm aerosol particles during clean and pollution periods. During the clean period, coinciding with the NPF event at approximately 10:00 (Fig. S6a), the number fraction of the MH mode for 50 nm particles increased (Fig. 4c). This suggests that intense photochemical reactions generate condensable species such as sulfuric, nitrate, and organics, which condense onto pre-existing aerosol particles, transforming them from externally mixed to internally mixed states (Wang et al., 2019). The MH mode was predominantly observed in the afternoon (14:00-16:00). After 16:00, the number fraction of NH mode particles increased, peaking at 20:00 (Fig. 4c and Fig. S7), primarily due to substantial traffic emissions during evening rush hour. During the pollution period, the $\kappa$-PDF of 50 nm aerosol particles exhibited a bimodal

distribution throughout the day. A higher number fraction of MH mode particles was observed in the early morning (Fig. 4d), likely due to intensified heterogeneous reactions of $NO_x$ during this period.

For 150 nm particles, the $\kappa$-PDF showed a similar diurnal variation pattern during both clean and pollution period, displaying a bimodal distribution throughout the day (Fig. 4e and 4f). During the clean period, the number fraction of MH for 150 nm particles was higher during the daytime than that at nighttime (Fig. 4e). As the sun rises, the elevation of the boundary layer allows more solar radiation to reach the ground, resulting in enhanced photochemical reactions and consequently an increase in aged particles, thereby enhancing hygroscopicity of particles and the number fraction of MH. Influenced by local primary emissions, the number fraction of NH increased after 18:00 (Fig. 4e). During the pollution period, the MH mode for 150 nm particles was stronger and more stable throughout the day compared to the clean period (Fig. 4f). The proportion of number fraction of NH consistently remained below 25 % throughout the day (Fig. S7). This could be attributed to the heterogeneous $NO_x$ reactions on the surface of aerosol particles, which significantly contribute to hygroscopicity by increasing the proportion of inorganic materials (Sun et al., 2016; Wang et al., 2016). AMS results show that the proportion of nitrate increased significantly during the pollution period (Fig. 8a), further supporting this viewpoint. The average nitrate mass fraction in $PM_1$ increased from 13 % ± 8 % before $P_{III}$ to 33 % ± 4 % during the $P_{III}$, indicating that the increase of nitrate is the main cause of heavy pollution in Beijing in autumn. Previous research in Beijing has shown that inorganic nitrate dominated the total nitrate mass, while organic nitrate accounted for approximately 10 % of the total nitrate mass during autumn (Xu et al., 2018). In recent years, with the implementation of strict air pollution control strategies, there has been a significant reduction in the concentrations of gaseous pollutants, especially $SO_2$ (Wang et al., 2022; Chu et al., 2020). During pollution events in Beijing, nitrates have replaced sulfates as the predominant inorganic component (Chen et al., 2019; Wang et al., 2019; Wang et al., 2022).

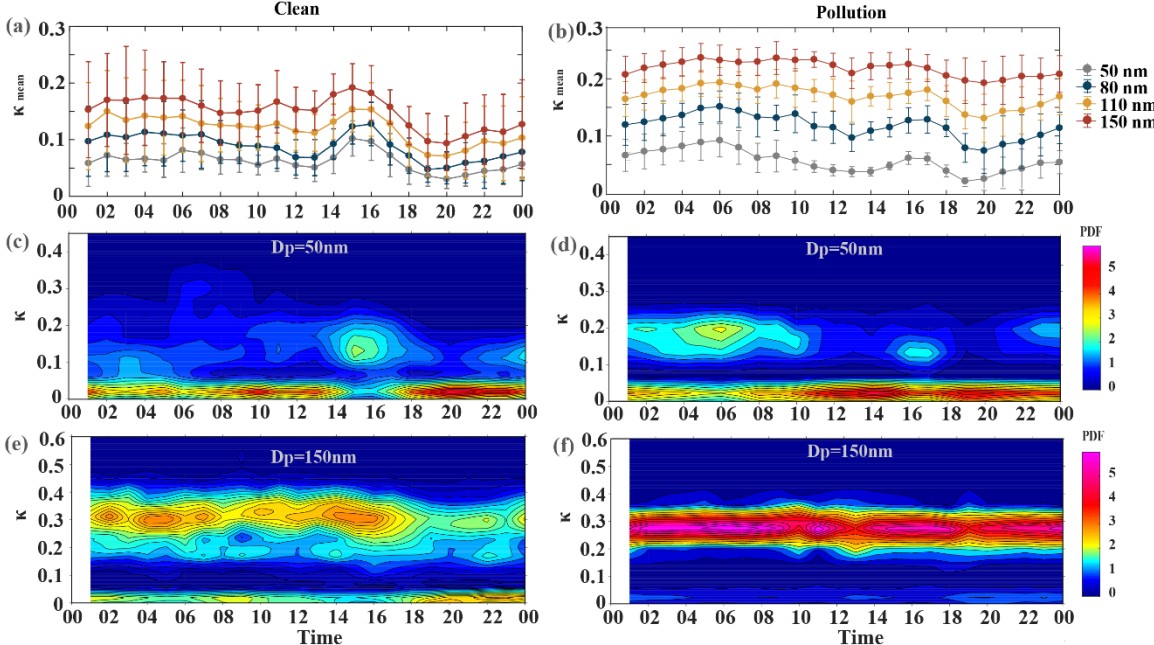

**Figure 4.** Diurnal variation of mean κ for different diameters and the error bars represent the standard deviations (a, b); κ-PDF for particles with $D_p$ = 50 nm (c, d) and κ-PDF for particles with $D_p$ = 150 nm (e, f) during the clean and pollution periods, respectively.

Figure 5a and 5b show the significant differences in the diurnal variation of $VSF_{mean}$ for particles with diameters ranging from 50 to 150 nm during two different periods. During the clean period, the highest $VSF_{mean}$ was observed in the morning (09:00-10:00), while the lowest was observed at midnight and in the afternoon. Conversely, during the pollution period, the highest $VSF_{mean}$ was observed in the afternoon (15:00-16:00) and the lowest observed in the morning for all particle sizes. During the clean period, particle volatility increased dramatically starting from 10:00 ($VSF_{mean}$ decreased) along with the occurrence of NPF events, indicating that the newly formed matter was more volatile. The $VSF_{mean}$ for 50 nm particles was noticeably lower than that for larger particles throughout the day. The volatility of 80 nm, 100 nm, and 150 nm particles reached peak around 14:00 (lowest VSF) and two hours later for 50 nm particles during the daytime. During the pollution period, the volatility of different diameters exhibited similar diurnal variation patterns (Fig. 5b). In addition, the variability in volatility for Aitken mode particles was greater than that for accumulation mode particles, consistent with findings from previous studies (Wang et al., 2017a; Zhang et al., 2016). The highest volatility for 50 nm and 80 nm particles occurred during the morning and evening rush hours, while it was lower in the afternoon and at night. This result was likely influenced by traffic emissions and meteorological conditions. The CAMS station, located approximately 200 m from a major road to the west, indicated significant influence from traffic emissions. The median diameter range of exhaust particles from gasoline vehicles is between 55-73 nm, with average diameter of 65 nm (Momenimovahed and Olfert, 2015). Thus, the volatility of 50 nm and 80 nm aerosol particles was particularly impacted by traffic emissions in

this study. Additionally, the lower boundary layer height and wind speed during the pollution period hindered the dispersion of primary emissions. Several studies have demonstrated that the volatility of organic compounds from traffic emissions decreases as the oxidation level increases (Huffman et al., 2009; Cao et al., 2018; Zhu et al., 2021; Yang et al., 2023). Therefore, the reduced volatility of particles in the afternoon and at night could be attributed to the increased oxidation degree of the organic components, leading to the production of more refractory organics (Cheung et al., 2016; Wang et al., 2017a; Wang et al., 2022).

Figure 5c-5f show the diurnal variation in VSF-PDFs for 50 nm and 150 nm particles during the clean and pollution periods, indicating that all VSF-PDFs exhibited both NV mode and VV mode. Notably, external mixing was more apparent during the night and early morning, especially for 150 nm particles during the clean period and 50 nm particles during the pollution period (Fig. 5d and 5e). This phenomenon could be attributed to the reduced boundary layer height, which leads to the accumulation of nonvolatile particulate matter emissions (e.g. BC, soot aggregates) from cooking or vehicles emissions. Diesel trucks are permitted to enter Beijing's 5th Ring Road from 00:00 to 06:00 (Beijing Municipal Bureau of Transportation, 2017), resulting in elevated nonvolatile particulate matter emissions during the night and early morning (Hakkarainen et al., 2023). As the boundary layer develops during the day, promoting vertical mixing, the number fraction of NV mode particles decreases. During the clean period, the number fraction of NV mode for 50 nm particles decreases around 10:00 (Fig. 5c), as these newly formed particles are primarily composed of volatile substances such as sulfates, ammonium, and organic matter during NPF days (Yue et al., 2010; Wu et al., 2016; Yang et al., 2021). During the pollution period, external mixing of 50 nm particles with volatile (e.g., organic compounds) and non-volatile matter primarily due to local primary emissions (Wehner et al., 2009; Wang et al., 2019).

The VSF-PDF of 150 nm aerosol particles during the pollution period exhibits a stronger and more stable VV mode, and a lower NV mode compared to the clean period (Fig. 5e and 5f). During the pollution period, primary emitted particles can be rapidly coated and aged through heterogeneous reactions (Hakala et al., 2016; Wang et al., 2016), causing the non-volatile particles to transition from an externally mixed mode with volatile particles to an internally mixed mode (Wehner et al., 2009; Yue et al., 2009; Hong et al., 2018). During the clean period, the active aging process in the afternoon promotes the mixing of primary particles with secondary particles, increasing the number fraction of VV mode for 150 nm particles. Influenced by primary emissions, the volatility of VV mode began to weaken after 18:00.

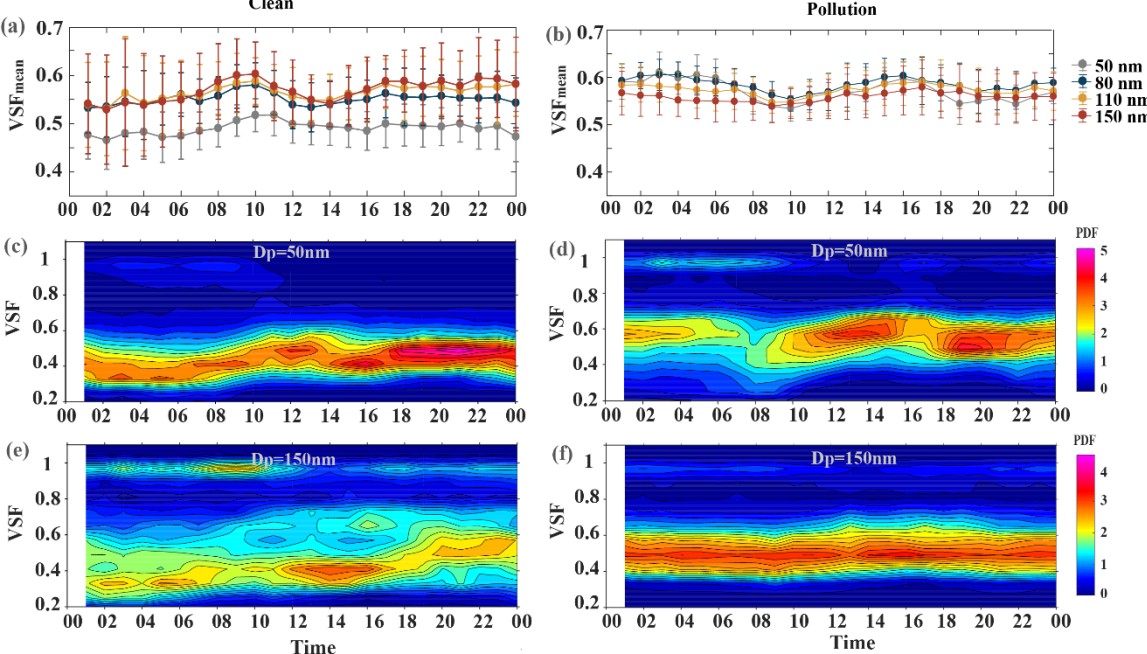

**Fig. 5.** Diurnal variation of mean VSF for different diameters and the error bars represent the standard deviations (a, b); VSF-PDF for particles with $D_p$ =50 nm (c, d) and VSF-PDF for particles with $D_p$ =150 nm (e, f) during the clean and pollution periods, respectively.

## 4.4 Size-resolved hygroscopicity and volatility during the clean and pollution periods

Figure 6 shows the statistics of size-resolved hygroscopicity and volatility, as well as the number fraction of nearly hydrophobic mode particles ($NF_{NH}$) and non-volatile mode particles ($NF_{NV}$) during the clean and pollution periods. As particle diameter increases, both the mean and median values of the HGF showed noticeable increases during both periods. Furthermore, for particles ranging from 80 nm to 150 nm, the hygroscopicity exhibited significant increases (p<0.0001) during the pollution period compared to the clean period (Fig. 6a). The mean HGF increased from 1.15 to 1.32 for particle sizes ranging from 50 nm to 150 nm during the clean period and from 1.15 to 1.44 during the pollution period (Table 1). In contrast to HGF, both the mean and median values of $NF_{NH}$ decreased with increasing particle size (Fig. 6b). The $NF_{NH}$ for particles ranging from 80 nm to 150 nm exhibited significant decreases (p < 0.0001) during the pollution period compared to the clean period. The mean $NF_{NH}$ decreased from 0.65 to 0.38 and from 0.67 to 0.14 for particle sizes ranging from 50 nm to 150 nm during the clean period and the pollution period, respectively (Table 1). Larger particles under the pollution period typically comprised more inorganic salts, enhancing their hygroscopicity. The variation in parameters for different particle sizes was much larger during the clean period than during the pollution period, as shown in Fig. 6. Additionally, we found more external mixed larger particles during the clean period, as indicated by larger $\sigma_{HGF\text{-}PDF}$ values (Table 1). Thus, the chemical composition may deviate significantly during the clean period. During the clean period, NPF events occur frequently, and larger particles are likely formed by the condensation of different substances (such as

sulfate, ammonium, and organics) on pre-existing particles, leading to a more complex chemical composition (Cheung et al., 2016; Wang et al., 2017a). However, during the pollution period, the larger particles are more aged

and contain a higher inorganic fraction, resulting the chemical composition more homogeneous (Wehner et al., 2009; Sun et al., 2016; Wu et al., 2016).

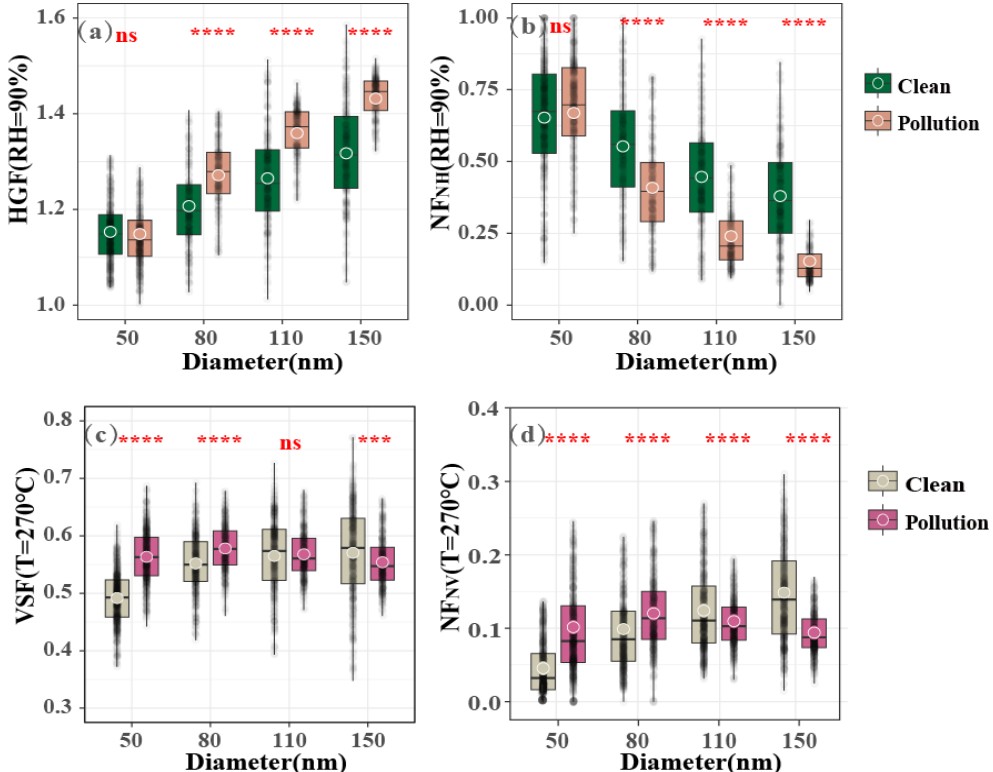

**Figure 6.** Boxplot of (a) size-resolved hygroscopic growth factor (HGF) and (b) the number fraction of nearly

hydrophobic mode particles ($NF_{NH}$) during the clean and pollution periods. (c) (d) same as (a) (b) but for volatile shrink factor (VSF) and number fraction of non-volatile mode particles ($NF_{NV}$), respectively. The two borders of box indicate the 25th and 75th percentiles, the line in box represents the median, whiskers indicate the 5th and 95th percentiles, and the circle in each box indicates the mean. Ns means the observed difference is not statistically significant, **** means P value ≤ 0.0001.

In terms of aerosol volatility, the mean and median values of VSF slightly increased with particle size during the clean period, indicating reduced volatility (higher VSF corresponds to lower volatility); However, no distinct trend was observed during the pollution period (Fig. 6c). The $NF_{NV}$ followed a similar trend to VSF (Fig. 6d). During the clean period, the mean VSF increased from 0.49 to 0.57, and $NF_{NV}$ increased from 0.05 to 0.15 for particles ranging from 50 nm to 150 nm. However, during the pollution period, the mean VSF was around 0.56 for all selected sizes,

with less than 5% variability among different particle sizes (Table 1). Notably, Aitken mode particles exhibited higher volatility and a lower $NF_{NV}$ during the clean period compared to the pollution period (Fig. 6c and 6d). This

difference was more pronounced for 50 nm particles compared to 80 nm particles. For accumulation mode particles, the situation was opposite, and the difference between two periods for 150nm particles was greater than for 110 nm particles. Previous research has shown that particles smaller than 100 nm exhibit lower $NF_{NV}$ on nucleation days than non-nucleation days, while results are reversed for particles larger than 100 nm (Wehner et al., 2009). During the clean period, nucleation events primarily contribute to the volatility of Aitken mode particles, while the growth of larger particles is mainly due to the coating effect of condensed vapors (primarily organic vapors) on pre-existing particles during NPF events (Wang et al., 2017a). Wu et al. (2017) suggest that during NPF events, the increase in non-volatile materials with particle size is caused by photochemically induced nucleation of condensing vapors or rapid chemical transformations occurring within the particle phase. The volatility observed in this study is higher than previous results from Beijing in 2015, where Wang et al. (2017) reported that the VSF of particles ranging from 40 nm to 150 nm was between 0.6 to 0.72 during the clean period and 0.65 during the pollution period. This increase in volatility could be attributed to the continuous reduction in BC concentration, which has decreased significantly from $6.25 \pm 5.73\ \mu g\ m^{-3}$ in 2012 to $1.80 \pm 1.54\ \mu g\ m^{-3}$ in 2020, and further to $0.79 \pm 0.63\ \mu g\ m^{-3}$ in 2022, due to the implementation of the Clean Air Action Plan in 2013 (Sun et al., 2022; Hu et al., 2024).

## 4.5 Relationship of hygroscopic and volatile properties

In order to understand the relationship between the hygroscopicity and volatility of particles, we investigated the correlation between the HGF and VSF, as well as $NF_{NH}$ and $NF_{LV}$, across different aerosol sizes during the clean and pollution periods. As shown in Fig. 7a and 7b, except for 50 nm particles, the VSF exhibited a decreasing trend with increasing HGF for all selected particle sizes during both periods, indicating that volatility increases with an increase in hygroscopicity. This trend suggests that the higher hygroscopic and volatile components within aerosol particles are predominantly inorganics. Furthermore, the correlations between the HGF and VSF were size-dependent, exhibiting stronger correlation and higher absolute slope values for larger particles, likely due to their higher proportion of inorganic components. Previous study has shown that as particle size increases, the inorganic mass fraction increases from approximately 35% to 70% (Wu et al., 2016). Another notable observation from Figures 7a and 7b is that the hygroscopicity and volatility of particles during the pollution period were less scattered compared to those during the clean period. This could be attributed to higher degree of aging and the more uniform chemical composition of particles during pollution period. However, 50 nm particles, which primarily consist of freshly emitted particulates, exhibited significant chemical composition variability due to contributions from different emission sources (Wang et al., 2019).

To further understand the relationship between hygroscopicity and volatility, we also investigated the correlation between $NF_{NH}$ and $NF_{NV}$ during the clean and pollution periods (Fig. 7c and7d). For submicron particles, those remaining non-volatile particles after being heated to 250-300 °C are generally considered nearly hydrophobic particles, primarily consisting of soot, carbonaceous particles and organic polymers in previous research (Cui et al 2021; Enroth et al., 2018; Massling et al., 2009; Wehner et al., 2009; Zhang et al., 2016; Mendes et al., 2018). Our result shows a weak correlation between $NF_{NH}$ and $NF_{NV}$, especially for Aitken mode particles. This finding is consistent with a previous study in Beijing (Wang et al., 2017a). However, the $R^2$ values from our study are lower than those for same diameter particles in the suburban Xiang He station. This discrepancy is influenced by presence of more soot and local light-absorbing carbonaceous aerosols in Xiang He, and these materials are neither hygroscopic nor volatile (Zhang et al., 2016). As shown in Fig.7c and 7d, all the points are below the 1:1 line, indicating that a certain proportion of nearly hydrophobic particles are volatile. $NF_{NV}$ for all particles were lower 0.4, $NF_{NH}$ for Aitken mode particles mainly ranged from 0.2 to 1.0, and for accumulation mode particles, it was focused below 0.8 and 0.4 during the clean and pollution periods, respectively. This indicates that $NF_{NH}$ and $NH_{NV}$ for accumulation mode particles were composed of similar components during the pollution period, likely BC and soot. A larger proportion of hydrophobic and volatile particles was found in the Aitken mode, which can be attributed to particles from NPF events and traffic emissions. These particles contain a variety of organic materials that are typically highly volatile but exhibit low hygroscopicity (Cai et al., 2022; Feng et al., 2023).

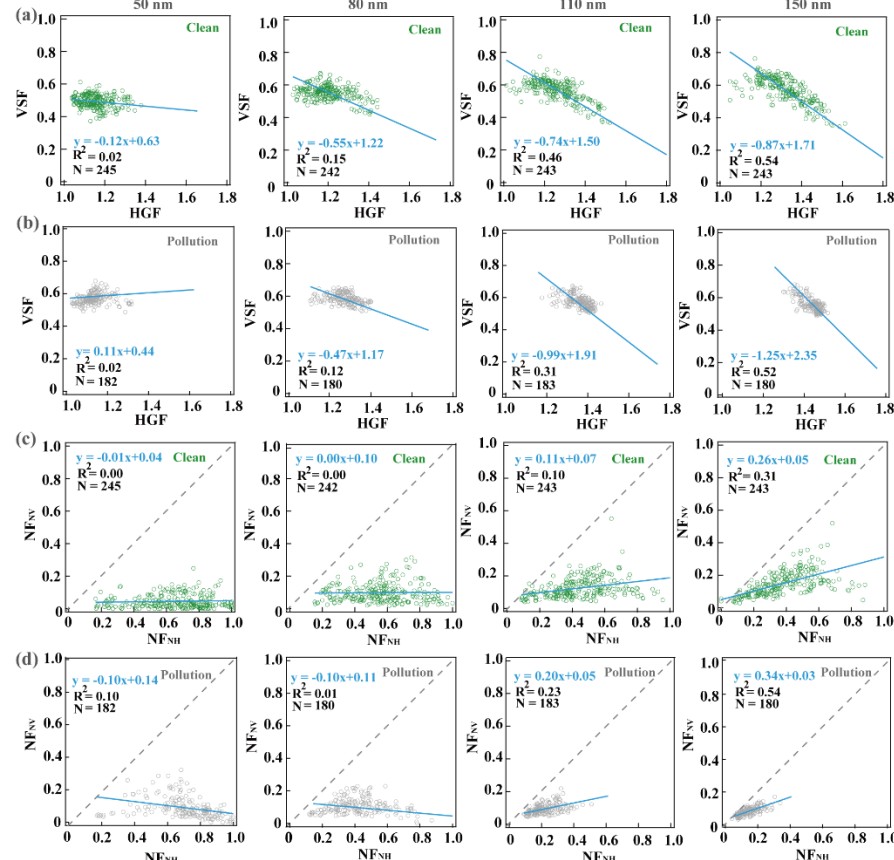

**Figure 7.** Relationship between the hygroscopic growth factors (HGFs) and the volatility shrink factors (VSFs) during the clean and pollution periods for different dry particle diameters (a, b); relationship between the $NF_{NH}$ and the $NF_{NV}$ during the clean and pollution periods (c, d) for different dry particle diameters.

## 4.6 Hygroscopicity of non-volatile core and volatile coating

For ambient aerosols, most compounds can volatilize below 270 °C (Enroth et al., 2018). However, certain compounds, such as BC, sea salt, and crustal materials, do not volatilize even at higher temperatures. Previous researches have shown that submicron atmospheric aerosols contain non-volatile components other than BC, possibly including organic polymers (Backman et al., 2010; Häkkinen et al., 2012; Hong et al., 2014). Furthermore, even newly nucleated particles retain a non-volatile core (Wehner et al., 2005; Ehn et al., 2007a; Wu et al., 2017). This section presents the measured hygroscopic growth of the non-volatile core after heated at 270 °C during the sampling period. Based on the simultaneously measured hygroscopic growth factor of particles and their non-volatile core, we also calculated the hygroscopic growth factor of volatile coatings under the assumption that the particles are a two-component system composed of a non-volatile core and a volatile coating. The VFR intuitively shows the variations of volume fraction of non-volatile core for different size particles (Fig. 8b). The mean VFR of particles ranging from 50 nm to 150 nm was $0.15 \pm 0.05$, $0.19 \pm 0.05$, $0.20 \pm 0.06$ and $0.20 \pm 0.07$, respectively. These results are slightly lower than those of an earlier study at urban site under the same heating temperature

(Enroth et al., 2018), which investigated the volatile properties of atmospheric aerosol particles (20–145 nm) and found a VFR of approximately 0.2-0.25. During the pollution period, especially during $P_{III}$, the VFR exhibits minimal variation. The mean $HGF_{coating}$ for particles ranging from 50 nm to 150 nm was $1.17 \pm 0.08$, $1.27 \pm 0.10$, $1.35 \pm 0.10$ and $1.41 \pm 0.10$, respectively, which was 2 % -7 % higher than mean HGF for corresponding particles size. As shown in Fig. 8c, variations in $HGF_{coating}$ basically similar to those hygroscopicity of unheated particles ($HGF_{mean}$) (Fig. S8) and exhibit significant size dependency. It is clear that the particle coating exhibits higher hygroscopicity during the pollution period. Figure 9 shows that $HGF_{coating}$ was influenced by the volume fraction of the coating in particles during the pollution period. The correlations between the $HGF_{coating}$ and the volume fraction of the coating in particles were stronger for the largest particles, suggesting that these particles were typically highly aged and had a higher proportion of inorganic components with a greater volume proportion of the coating. AMS data also show that the relative mass fraction of inorganic components in the particulate phase significantly increases during the pollution period (Fig. 8a). The nitrate content dominated the inorganic components of aerosol particles during the pollution period, reaching 33 % during the $P_{III}$ and likely being the major contributor to the increase in $HGF_{coating}$.

The mean $HGF_{core}$ for particles ranging from 50 nm to150 nm in this study was $1.08 \pm 0.03$, $1.07 \pm 0.03$, $1.07 \pm 0.03$, and $1.09 \pm 0.04$, respectively. Unlike $HGF_{coating}$, $HGF_{core}$ showed no size-dependent variation, likely due to the similar proportions of residual hygroscopic components in particles across different sizes (Fig. 8d). Given that BC and mineral dust are not hygroscopic, the non-volatility fraction of particles still contains other hygroscopic components, likely highly oxidized organic polymers (Liu et al., 2021; Hong et al., 2014). Studies on forest aerosols have shown that the mass fraction of non-BC material accounts for 13 %-45 % of the total remaining mass after heated at 280 °C, correlating significantly with mass fraction of organic nitrates (Häkkinen et al., 2012). The organic nitrates were effectively quantified to contribute a notable fraction to organic aerosol (OA), namely 9 %-20 % in autumn (Yu et al., 2019). Additionally, such non-volatile organic compounds may also be related to humic-like substances (Wu et al., 2009), which are significant components of fine particulate matter in Beijing (Ma et al., 2018). These oxygenated organic aerosols do not volatilize even after heated to 270 °C, remaining as part of the non-volatile components contributing to the core's hygroscopicity.

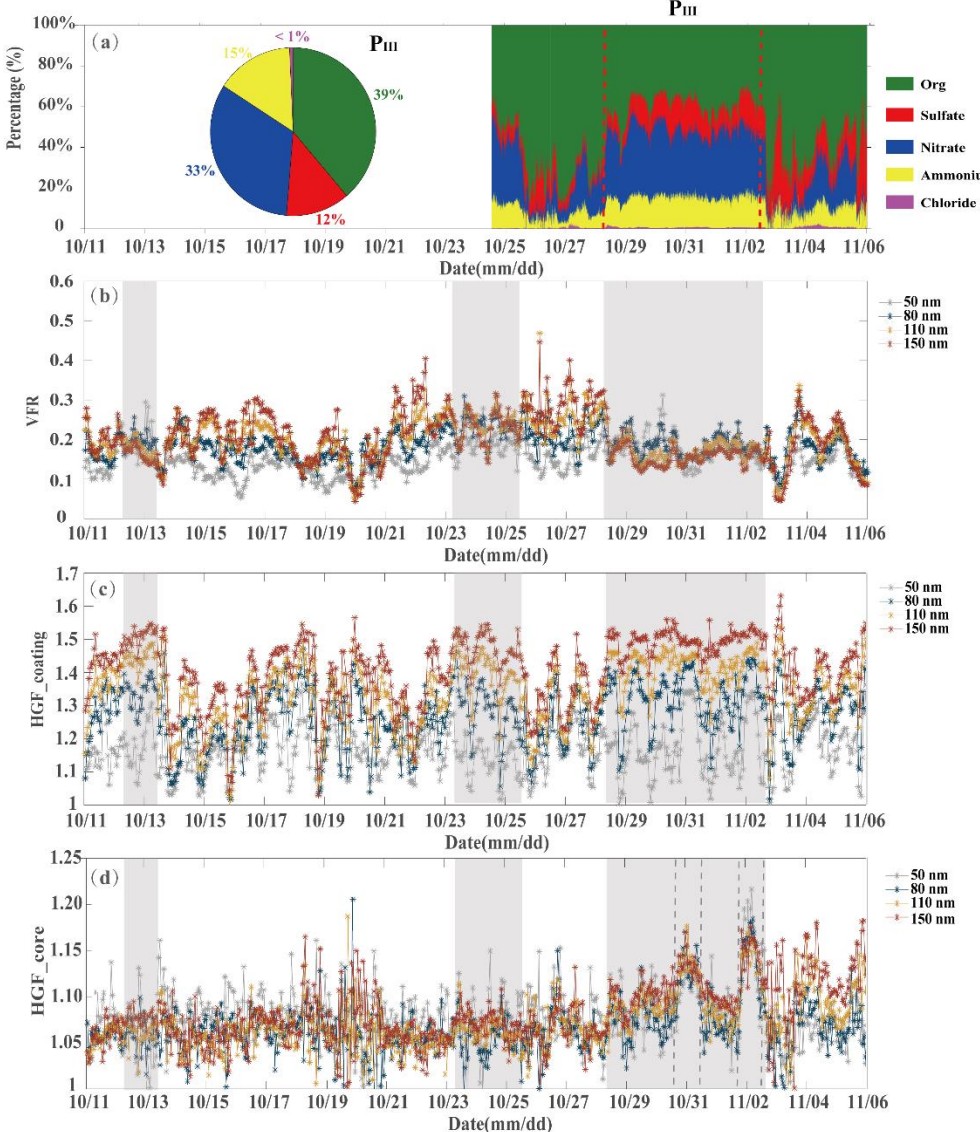

**Figure 8**. Time series of the main mass fraction of chemical components in PM$_1$ during the sampling period based on the available AMS data (a), The volume fraction of non-volatile core (VFR) (b), calculated HGF$_{coating}$ (c) and measured HGF$_{core}$ (d) for particles from 50 nm to 150 nm during the sampling period. The shadows represent the pollution period and the dashed lines represent the period during which the hygroscopicity of non-volatile core was relatively high.

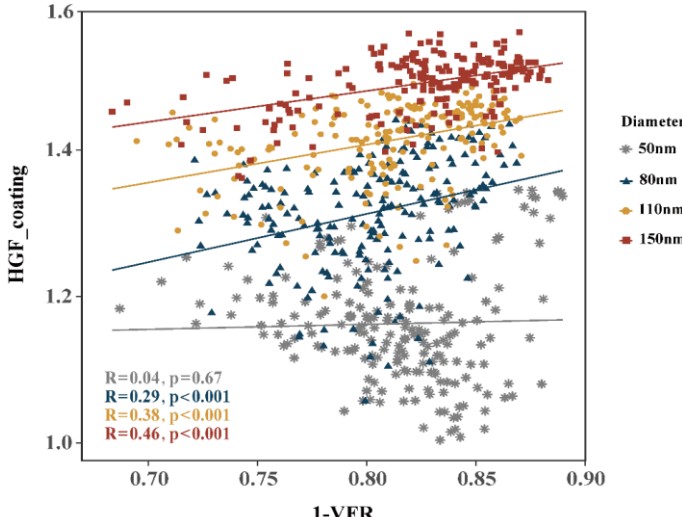

**Figure 9.** Correlations between the hygroscopic growth factor of volatile coating and its volume fraction for different particle sizes during the pollution period.

Figure 8d shows two significant increases in the hygroscopicity of the non-volatile core during periods of severe pollution. The first increase, referred to as event I, occurred from 30 October, 15:00 to 31 October, 15:00. The second, event II, occurred from 1 November, 17:00 to 2 November, 17:00. To further investigate these events, we analyzed the 48 h backward trajectories reaching the sampling site at a height of 500 m above ground level (Fig. 10). During event I, the air mass originated from Shanxi (Fig. 10a). Four hours later, the air mass passed through Shandong, Bohai Sea, and Tianjin before reaching the sampling site, eventually shifting to northwest Mongolian at the end of event I. The early stage of event II was initiated by a short-distance air mass from Hebei (Fig. 10b). Subsequently, the air mass shifted to the Bohai Sea, Shanxi, and then transitioned into a long-range, clean air mass from the northwest at the end of event II. During these two events, part of air masses reached the sampling site either passing through or originating from the Bohai Sea. Moreover, these air masses traveled below 1 km at lower speeds, enhancing the possibility of marine aerosol entering the atmosphere (Fig. 10c and 10d). Thus, the observed increases in the hygroscopicity of non-volatile core particles were probably influenced by marine aerosol. To investigate the enhanced hygroscopicity of the non-volatile core, we further analyzed the mean VSF-PDF and mean VHGF-PDF during the periods when air masses reached the sampling site both by passing through and originating from the Bohai Sea.

Figure 11 displays the mean VSF-PDF and mean VHGF-PDF for 150 nm particles during the periods when the air mass passed through (30 October, 21:00 to 31 October, 03:00 in event I) and originated from the Bohai Sea (2 November, 01:00 to 05:00 in event II), respectively. The top axis of the plot denotes the particle diameter. The VSF-PDF in Fig. 11 shows a minor non-volatile (NV) mode and a major very volatile (VV) mode (marked by $D_p$ (270 °C, RH$_{dry}$)), while the VHGF-PDF displaced three modes, marked by $D_{p1}$ (270 °C, 90 % RH), $D_{p2}$ (270 °C, 90 %

RH), and $D_{p3}$ (270 °C, 90 % RH), respectively. Since the NV mode was smaller than the $D_{p3}$ (270 °C, 90 % RH) mode, we assume the three modes in the VHGF-PDF result from the hygroscopic growth of VV mode particles after heating. During the air mass passed through the Bohai Sea in event I (Fig. 11a), the particle size at the peak of the VV mode in the VSF-PDF was 75 nm ($D_p$ (270 °C, $RH_{dry}$)). After hygroscopic growth, the peaks of the three modes

in the VHGF-PDF were observed at $D_{p1}$ (270 °C, 90% RH) of 77 nm, $D_{p2}$ (270 °C, 90% RH) of 96 nm, and $D_{p3}$ (270 °C, 90% RH) of 135 nm. Therefore, the hygroscopic growth factor of the hygroscopic components in the non-volatile core was calculated as follows: $D_{p1} / D_p = 77/75 = 1.03$, $D_{p2} / D_p = 96/75 = 1.28$, and $D_{p3} / D_p = 135/75 = 1.80$. When the air mass originated from the Bohai Sea during event II (Fig. 11b), the particle size at the peak of the VV mode in the VSF-PDF was 77 nm ($D_p$ (270 °C, $RH_{dry}$)), the VHGF-PDF also exhibited a trimodal distribution

after rehydration. Same calculation method as above, the HGFs corresponding to the three peaks were 1.21, 1.41 and 1.81, respectively. While this estimate carries some uncertainty, it provides valuable insights into the complexity of aerosol hygroscopicity and volatilization.

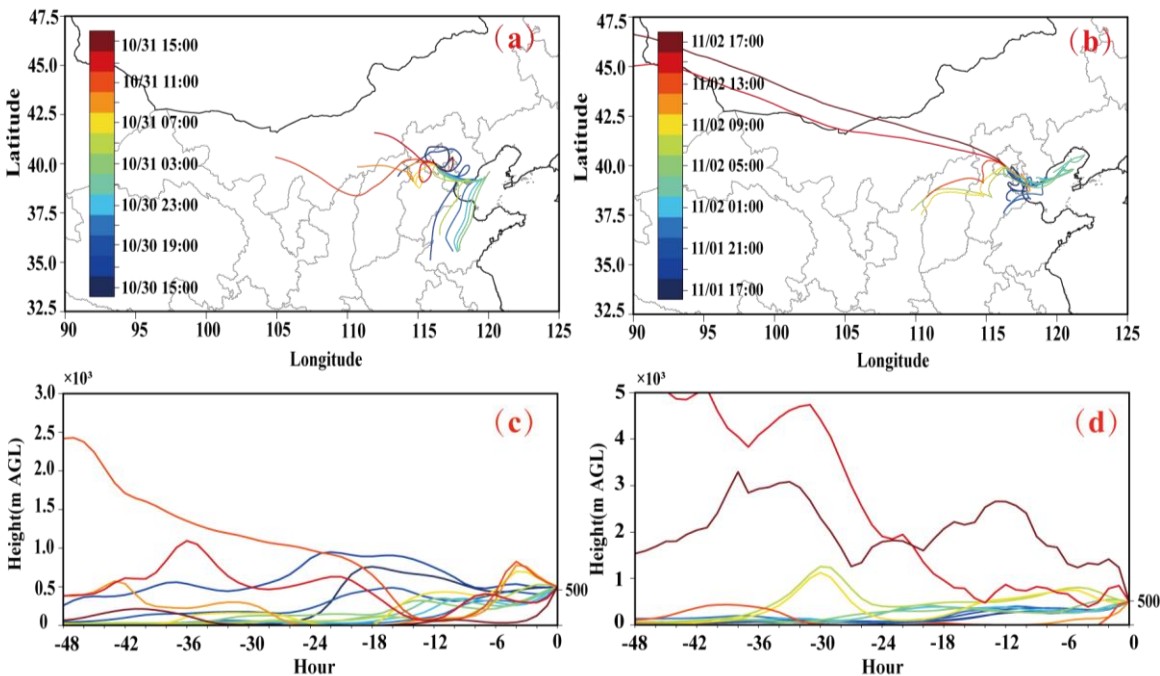

**Figure 10.** The 48-h back trajectories arriving at the height of 500 m AGL at the CAMS site were calculated every

two hours from 30 October, 15:00 to 31 October, 15:00 including event I (a), and from 1 November, 17:00 to 2 November, 17:00 including event II (b). (c) and (d) show the air mass heights of the backward trajectories corresponding to (a) and (b), respectively. The color scale represents the time of air mass backward trajectory arrived at the sampling site.

Sea-spray aerosol (SSA) consists of particles directly produced at the sea surface and suspended in the air, with

sizes ranging from less than 10 nm to several millimeters (Bates et al., 2012; De Leeuw et al., 2011), and SSA can

even stay in the atmosphere for days (Grythe et al., 2014). Sub-100 nm SSA non-volatile components at temperatures 250-550 °C include sea salt and non-sea salt components, which could be organic fraction, biological materials, BC, and minerals (Xu et al., 2020; Mallet et al., 2016; Cravigan et al., 2015). Even when heated at 550 °C, 29 % to 48 % of nonvolatile organic fraction still remains in 60 nm SSA (Mallet et al., 2016). The HGFs of SSA

ranging from 1.0 to 1.1, typically composed of biological organic components or anthropogenic emissions such as BC (Kim et al., 2015), while HGFs between approximately 1.1 and 1.4 are associated with organics aerosols (Gantt and Meskhidze, 2013; Ovadnevaite et al., 2011; Asmi et al., 2010). In the Aitken mode, the HGF of 1.8 in non-volatile marine particles is likely due to the presence of sea salt (Sellegri et al., 2008). In summary, the non-volatile SSA from these two events likely comprise a mixture of sea salt, non-volatile organic compounds, and BC. Before

reaching the sampling site, SSA particles in the air masses were further aged and grown during transport, forming a volatile coating on their surface that reduced their hygroscopic properties. Therefore, no significant increase in particle hygroscopicity was found before heating (Fig. S8). However, this study cannot definitively identify the specific components of marine aerosols transported by the air masses due to available measurements.

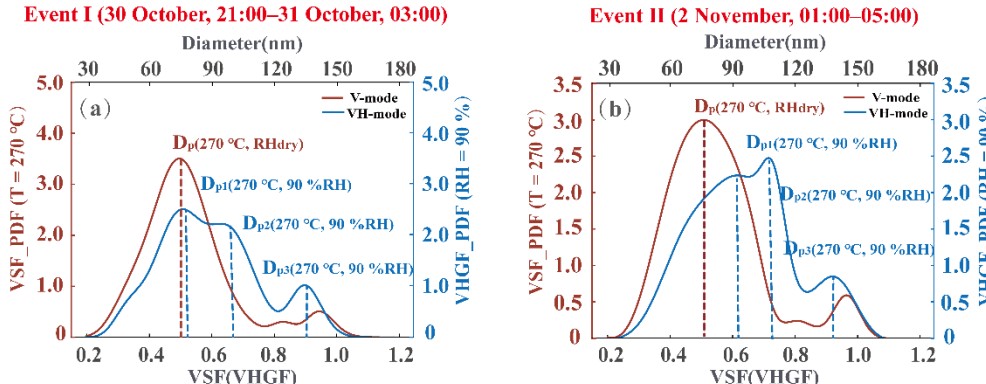

**Figure 11.** Representative case studies of VH-TDMA measurements. The mean VSF-PDF and mean VHGF-PDF of 150 nm particles from 30 October, 21:00 to 31 October, 03:00 in event I (11a), and form 2 November, 01:00 to 05:00 in event II (11b).

## 5 Conclusions

The Volatility Hygroscopicity Tandem Differential Mobility Analyzer (VH-TDMA) was used to measure the

hygroscopicity and volatility of the aerosol particles (50 nm, 80 nm, 110 nm, and 150 nm) in Beijing from 11 October to 6 November 2023. The HGF-PDF and VSF-PDF for different particle sizes displayed a bimodal distribution, indicating a clear state of external mixing throughout the study. The mean HGF for particles from 50 to 150 nm during the sampling period was $1.15 \pm 0.07$, $1.24 \pm 0.08$, $1.30 \pm 0.09$, and $1.36 \pm 0.10$, respectively, while the mean VSF was $0.51 \pm 0.05$, $0.55 \pm 0.04$, $0.56 \pm 0.05$, and $0.56 \pm 0.07$, respectively. To better compare the

hygroscopic and volatile properties of particles under different weather conditions, the sampling period was divided

into clean periods and pollution periods. The air mass arriving at the sampling site was mainly from the northwest and the south during the clean period and pollution period, respectively. Hygroscopicity increased with particle size in both clean and pollution periods, with a more pronounced increase during the pollution period, where mean HGF rose from 1.15 to 1.44, compared to 1.15 to 1.32 during the clean period. In contrast, the mean $NF_{NH}$ decreased with an increase in particle size, from 0.65 to 0.38 during the clean period and from 0.67 to 0.14 during the pollution period. The average nitrate mass fraction in $PM_1$ increased from 13 % ± 8 % before $P_{III}$ to 33 % ± 4 % during $P_{III}$, reflecting the increased hygroscopicity of larger particles could be attributed to their greater age and higher proportion of inorganic component. Volatility decreased and $NF_{NV}$ increased slightly with increase of particle size during the clean period, no apparent trend was observed for the pollution period.

The $\kappa_{mean}$ exhibited similar diurnal variations during both clean and pollution periods. However, it peaked in the late afternoon during the clean period and in the early morning during the pollution period. This result could be due to the dominance of photochemical reactions during the daytime in the clean period and stronger heterogeneous reactions at night during the polluted period. The mean VSF displayed the different diurnal variations during the clean and pollution periods. The highest VSF was observed in the morning, while the lowest was observed at midnight and in the afternoon for particles during the clean period. Conversely, the highest VSF in the afternoon with the lowest observed in the morning during the pollution period. During the clean period, increased volatility was observed during daytime due to NPF events, driven by the formation of volatile matter and the coating effect of condensable vapors. During the pollution period, the reduced volatility of particles in the afternoon could be attributed to an enhanced oxidation degree of organic components. Accumulation mode particles exhibited a stronger and less variable VV and MH modes, due to the particles get aged and mixed with newly emitted particles during pollution period. Additionally, a positive correlation was identified between hygroscopicity and volatility, as well as between the number fraction of nearly hydrophobic and non-volatile particles during both the clean and pollution periods.

Furthermore, the study calculated the $HGF_{core}$ and $HGF_{coating}$ of particles after heating. The mean $HGF_{coating}$ for particles ranging from 50 nm to 150 nm was 1.17 ± 0.08, 1.27 ± 0.10, 1.35 ± 0.10 and 1.41 ± 0.10, respectively, which is 2 %–7 % higher than the mean HGF for similarly sized particles. The mean $HGF_{core}$ ranging from 50 nm to 150 nm is 1.08 ± 0.03, 1.07 ± 0.03, 1.07 ± 0.03 and 1.09 ± 0.04, respectively. The $HGF_{core}$ values were influenced by marine aerosols during air mass passages and originating from the Bohai Sea. These findings provide valuable scientific data for comprehensively understanding the hygroscopic and volatile properties of urban aerosols in Beijing. In the future, obtaining information on the hygroscopic properties of low-volatile organics will be important

and will aid in interpreting the VH-TDMA results, thus improving our understanding of the chemical properties of submicron aerosol. Additionally, investigating the hygroscopicity of aerosol particles heated at different temperatures and locations will further elucidate the structure and formation mechanisms of aerosol particles.

## Data availability.

The data in the study are available from the corresponding author upon request (jysun@cma.gov.cn).

## Competing interests.

The authors declare that they have no conflict of interest.

## Author contributions.

AYY analyzed the observational data and prepared the figures of the manuscript. JYS designed the study and

outlined the manuscript. JYS, AYY and JYL performed the instrument deployment and operation. All co-authors discussed the results and commented on the manuscript. AYY prepared the manuscript with contributions from all authors.

## Acknowledgments.

This research was supported by supported by the National Key R&D Program of China (grant no.

2023YFC3706305), China Meteorological Administration (CXFZ2024J039), National Natural Science Foundation of China (42075082, 42275121), Chinese Academy of Meteorological Sciences (2023Z012, 2022KJ002, 2024Z006), and Innovation Team for Haze-fog Observation and Forecasts of MOST.

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
