# Peer review of "Size-resolved hygroscopicity and volatility properties of ambient urban aerosol particles measured by the VH-TDMA system in Beijing"

_EGUsphere, 2024_

## Referee Comment (RC2)

Report

Yu et al., describes an interesting study on the hygroscopicity and volatility of aerosols in Beijing urban areas during four weeks' field campaign using a Volatility Hygroscopicity Tandem Differential Mobility Analyzer (VH-TDMA) system, a twin scanning mobility particle sizer, a high-resolution time-of-flight aerosol mass spectrometer and other meteorological instruments. The volatile shrink factor (VSF) and the hygroscopic growth factor (HGF) are reported for aerosols having different sizes: 50 nm, 80nm, 110 nm and 150 nm. Furthermore, the authors try to display the size-resolved probability function on the HGF and VSF determined by two Scanning Mobility Particle sizers (SMPS) implicated in the VH-TDMA system. The bimodal distribution has been shown for both two probability functions. During the field campaign, one clean period and three pollution periods are identified. The authors report also the relation between the hygroscopicity and volatility for submicron aerosols. The back trajectories are also carried out for this observation site. According to the bimodal distribution of size-resolved HGF and VSF, the authors suggest that the presence of submicron aerosols in Beijing urban areas is probably under the external mixture due to different sources.

**The manuscript brings a comparatively interesting report but the presentation way, the data analysis methodology and all results related to the multi-charge effect (including HGF-PDFs, VSF-PDFs ,kappa and relation between hygroscopicity and volatility) need to be carefully reviewed before the submission.**

**1st major comment:** the presentation of the measurement campaign is insufficient. A general configuration of all instruments need to be well described, especially how the aerosols are sampled for different instruments. Only a short presentation has been done to indicate the measurement period and the sampling location. All instruments and data should be summarized in this section. However, several fragments could be found in the section of results and discussions, such as meteorological information, back trajectories analysis, etc. Furthermore, the instrumental information on the three DMAs and two CPCs implicated in the VH-TDMA system is missing, i.e., the model, working conditions... The authors really need to re-organize this section.

**2nd major comment:** an important part of the conclusions of this manuscript is based on the bimodal distribution of size-resolved HGF and VSF. However, the phenomenon of multicharged effect on the incoming aerosols by the neutralizer is well known. The authors do not provide the size distribution of selected particles by the DMA1 but only suppose that selected particles are monodispersed. The double charged particles potentially correspond to the larger particles which could contribute to the second mode of HGF and VSF probability functions, i.e., it is not clear whether the second mode is contributed by the different hygroscopic or volatile compounds in the aerosols or is contributed to the double charged particles selected by the DMA (generally these double charged particles are larger than what is commanded in the DMA selection). According to the literature, the double charge effect influences importantly the hygroscopicity analysis of SOA (Bouzidi et al., 2022), combustion aerosols (Petters et al., 2009;

Wu et al., 2020), oxygenated aerosols (Petters et al., 2007) at the laboratory and also the measurement during the field campaign (Mochida et al., 2010). Kim et al., 2023 reported a influence of 20% on the $k_{CCN}$. Several studies suggest how to minimize the double charge effect on the CCN counter (Wang et al., 2015) and HTDMA (Barrett et al., 2012; Duplissy et al., 2009; Oxford et al., 2022). The correction of this multiple charge effect is also provided by Kim et al., 2023 and Petters, 2018. The authors need to take into account this effect in order to better illustrate the conclusions.

Comments in details:

Line 88, what is the reason that authors chose 270 °C for studying the hygroscopicity of non-volatility particles while this temperature is usually set as 300 °C in the literature?

Line 89, in the section *"experimental set-up"*, the authors give a description of each instrument used in this study but no information about how aerosols are sampled among different instruments, that is extremely important to validate the results. For example, were these instruments connected to a common sampling inlet or separately? It is suggested to introduce globally the campaign at the very beginning of this section, including the sampling site, the instruments, sampling conditions and the sources of all meteorological data.

Line 97, can authors provide a reference which gives more detailed information on this VH-TDMA (TROPOS, Germany)? Otherwise, authors should provide them in this manuscript. For instance, the model of dryer is missing and at what RH aerosols enter in the neutralizer?

Line 98, what is the model of the DMA1 and in which conditions it works, for example the sheath flow rate? Same questions for the DMA2 and DMA3.

Line 98, what is the model of two CPCs and what size range they measure?

Line 151 and line 156, authors need to declare clearly how the *Dp (Troom, RHdry)* is defined for the data treatment in this manuscript.

Line 161, for having a size-resolved HGF-PDF, how the *Dp (Troom, RHdry)* is defined, the size that the DMA1 selected or a probability density distribution of "monodisperse" aerosols selected by the DMA1? In both case, the multi-charge effect is non-negligible. It is necessaire to provide the size distribution of aerosols selected by the DMA1 or to consider the size distribution of sampled aerosols from site to evaluate this multicharged effect (Duplissy et al., 2009; Kim et al., 2023).

Line 165, the same question as the previous. How the *Dp (Troom, RHdry)* is defined for having a VSF-PDF?

Line 165, the method how to provide the *VHGF-PDF* is missing. Since authors have observed the binormal distribution of the non-volatile core of aerosols after heated, how *Dp(270 ℃ ,RHdry)* is defined for obtaining your results (for example the VHGF-PDF in figure 11), using the first mode size or the second mode size or the probability density distribution?

Line 183, authors really need to take into account the multi-charge effect before calculating the $\sigma HGF\text{-}PDF$.

Line 196, could authors provide the size distribution of aerosols measured by the PNSD to show the seven NPF events?

Line 209, authors need to take into account the multi-charge effect before giving the conclusion.

Line 216, in the case of non-consideration of multi-charge effect, the variation (number and mode size) of ambient aerosol size distribution due to clean and/or pollution events could contribute to the variations in the HGF-PDFs and VSF-PDFs. It is suggested that authors provide not only the mean particle size distribution during the clean period as Figure S4, but also that as function as time during the whole measurement campaign.

Line 240, "*narrower*" is not a quantitatively description. Coud authors provide the geometrical standard deviation of two modes for different cases in order to illustrate this conclusion?

Line 265, figure 3 shows the mean HGF-PDF and the mean VSF-PDF of particles having different sizes. In figure 3 a, how to explain the HGF of left part in the first mode (around ¼ visibly) is lower than 100%?

Line 268, the general presentation of the table 1 is missing. In the table 1, authors need to well define the "*Total*". Does "Total" means the whole period from 11 October to 6 November 2023? If yes, it is interesting that the mean k of 50 nm in "*Total*" is slightly higher than that in both "*Clean*" and "*Pollution*". Is it significant large in the period which is not "*Clean*" and "*Pollution*"?

Line 314, "*AMS results show that nitrate is the main inorganic component of PM1 (Fig. 8a), further supporting this viewpoint.*" It is known that nitrate signals in the AMS could represent inorganic nitrate compounds or organic nitrate compounds. Graeffe et al., 2023 reported how to estimate the organic nitrate. Authors should provide more information such as NO+/NO2+ ratio to support this conclusion.

Line 329, "*During the clean period, particle volatility increased dramatically around 10:00 LT (VSFmean decreased) along with the occurrence of NPF events, indicating that the newly formed matter was more volatile.*" Replace "*around*" by "starting from".

Line 348, "*Notably, external mixing was more apparent during the night and early morning, especially for 150 nm particles during the clean period and 350 50 nm particles during the pollution period (Fig. 5d and 5e). This phenomenon could be attributed to the reduced boundary layer height, which leads to the accumulation of nonvolatile particulate matter emissions (e.g. BC, soot aggregates) from cooking or vehicles emissions.*" Authors need to take into account the multi-charge effect which could contribute to the VSF-PDF the same way as the external mixing. It is suggested to combine off-line technics such as Transmission electron microscopy to have a direct prove on the presence of soot particles and/or sea salts mentioned later.

Line 467, it is suggested to present ZSR relation and the method of calculating HGF$_{coating}$ in the section of "*data analysis*".

Line 479, "*As shown in Fig. 8c, variations in HGFcoating basically similar to those hygroscopicity of unheated particles (HGFmean) (Fig.S7) and exhibit significant size dependency*" What is the difference between figure.S7 and figure.2 and figure.S2 on the part of HGF? What is the interest to show the same data by using size independent time series (figure.S7) and using size-resolved probability function time series (figure.2 and figure.S2)? What is the procedure authors calculate HGFcoating, by using HGFmean or HGF-PDF or some other method for the core and for the original aerosols?

Line 501, according to the figure 11, the distribution of core particles (after the heating of 270 °C) is not monodisperse. How authors calculate HGF-core using equation 3?

Line 522, it is the first time to see "VHGF-PDF". The explanation is required.

Line 535, the time scale with multi colors is not explained.

Barrett, S. R. H., Yim, S. H. L., Gilmore, C. K., Murray, L. T., Kuhn, S. R., Tai, A. P. K., Yantosca, R. M., Byun, D. W., Ngan, F., Li, X., Levy, J. I., Ashok, A., Koo, J., Wong, H. M., Dessens, O., Balasubramanian, S., Fleming, G. G., Pearlson, M. N., Wollersheim, C., ... Waitz, I. A. (2012). Public Health, Climate, and Economic Impacts of Desulfurizing Jet Fuel. *Environmental Science & Technology*, *46*(8), 4275–4282. https://doi.org/10.1021/es203325a

Bouzidi, H., Fayad, L., Coeur, C., Houzel, N., Petitprez, D., Faccinetto, A., Wu, J., Tomas, A., Ondráček, J., Schwarz, J., Ždímal, V., & Zuend, A. (2022). Hygroscopic growth and CCN activity of secondary organic aerosol produced from dark ozonolysis of γ-terpinene. *Science of The Total Environment*, *817*, 153010. https://doi.org/10.1016/j.scitotenv.2022.153010

Duplissy, J., Gysel, M., Sjogren, S., Meyer, N., Good, N., Kammermann, L., Michaud, V., Weigel, R., Martins dos Santos, S., Gruening, C., Villani, P., Laj, P., Sellegri, K., Metzger, A., McFiggans, G. B., Wehrle, G., Richter, R., Dommen, J., Ristovski, Z., ... Weingartner,

E. (2009). Intercomparison study of six HTDMAs: Results and recommendations. *Atmospheric Measurement Techniques*, *2*(2), 363–378. https://doi.org/10.5194/amt-2-363-2009

Graeffe, F., Heikkinen, L., Garmash, O., Äijälä, M., Allan, J., Feron, A., Cirtog, M., Petit, J.-E., Bonnaire, N., Lambe, A., Favez, O., Albinet, A., Williams, L. R., & Ehn, M. (2023). Detecting and Characterizing Particulate Organic Nitrates with an Aerodyne Long-ToF Aerosol Mass Spectrometer. *ACS Earth and Space Chemistry*, *7*(1), 230–242. https://doi.org/10.1021/acsearthspacechem.2c00314

Kim, N., Su, H., Ma, N., Pöschl, U., & Cheng, Y. (2023). A multiple-charging correction algorithm for a broad-supersaturation scanning cloud condensation nuclei (BS2-CCN) system. *Atmospheric Measurement Techniques*, *16*(11), 2771–2780. https://doi.org/10.5194/amt-16-2771-2023

Mochida, M., Nishita-Hara, C., Kitamori, Y., Aggarwal, S. G., Kawamura, K., Miura, K., & Takami, A. (2010). Size-segregated measurements of cloud condensation nucleus activity and hygroscopic growth for aerosols at Cape Hedo, Japan, in spring 2008. *Journal of Geophysical Research: Atmospheres*, *115*(D21). https://doi.org/10.1029/2009JD013216

Oxford, C. R., Chakrabarty, R. K., & Williams, B. J. (2022). Evaluation of assumptions made by Hygroscopic Tandem Differential Mobility Analyzer inversion routines. *Journal of Aerosol Science*, *162*, 105955. https://doi.org/10.1016/j.jaerosci.2022.105955

Petters, M. D. (2018). A language to simplify computation of differential mobility analyzer response functions. *Aerosol Science and Technology*, *52*(12), 1437–1451. https://doi.org/10.1080/02786826.2018.1530724

Petters, M. D., Carrico, C. M., Kreidenweis, S. M., Prenni, A. J., DeMott, P. J., Collett Jr., J. L., & Moosmüller, H. (2009). Cloud condensation nucleation activity of biomass burning aerosol. *Journal of Geophysical Research: Atmospheres*, *114*(D22). https://doi.org/10.1029/2009JD012353

Petters, M. D., Prenni, A. J., Kreidenweis, S. M., & DeMott, P. J. (2007). On Measuring the Critical Diameter of Cloud Condensation Nuclei Using Mobility Selected Aerosol. *Aerosol Science and Technology*, *41*(10), 907–913. https://doi.org/10.1080/02786820701557214

Wang, Z., Su, H., Wang, X., Ma, N., Wiedensohler, A., Pöschl, U., & Cheng, Y. (2015). Scanning supersaturation condensation particle counter applied as a nano-CCN counter for size-resolved analysis of the hygroscopicity and chemical composition of nanoparticles. *Atmospheric Measurement Techniques*, *8*(5), 2161–2172. https://doi.org/10.5194/amt-8-2161-2015

Wu, J., Faccinetto, A., Grimonprez, S., Batut, S., Yon, J., Desgroux, P., & Petitprez, D. (2020). Influence of the dry aerosol particle size distribution and morphology on the cloud condensation nuclei activation. An experimental and theoretical investigation. *Atmospheric Chemistry and Physics*, *20*(7), 4209–4225. https://doi.org/10.5194/acp-20-4209-2020

---

## Author Comment (AC1)

**Response to Reviewers' comments**

We thank the reviewers for their thoughtful and constructive comments that help us improve the manuscript substantially. We have revised the manuscript accordingly. Listed below is our point-to-point response in blue to each comment that was offered by the reviewers. We hope that our revised manuscript will now be suitable for publication in ACP.

**Response to Reviewer #1**

The manuscript is presents volatility and hygroscopicity data from urban ambient measurements in a clear manner and with good scientific quality. I would like to propose that this manuscript would be accepted with minor revisions. The first comment is related to the volatility measurement temperature that was chosen to be 300 Celsius. All the references given in the introduction are from measurements using 270 Celsius temperature. What lead to this decision? How to compare results with pre-existing data I there is a clear difference in the temperature used? Please add some text about this issue / decision. Second comment is about the experimental setup used: did you use $PM_{2.5}$ cut for the sampling, or a cyclone?

**R:** The authors thank the reviewer's positive comments. The heating temperature of 270 °C for ambient aerosol particles was selected based on the volatility characteristics of relevant inorganic salts, organic compounds, black carbon, soot, and sea salt. At this temperature, atmospheric sulfates, nitrates, and most organic compounds are typically volatile, while soot, sea salt, mineral dust and certain organic polymers remain refractory (Enroth et al., 2018; Johnson et al., 2005). Previous studies have shown that the major inorganic compounds in the atmosphere, such as ammonium nitrate and ammonium chloride, completely volatilize below 150 °C, and ammonium sulfate volatilizes completely between 180 °C and 230 °C (Huffman et al., 2008; Feng et al., 2023). Most organic compounds evaporate at relatively lower temperatures (Tritscher et al., 2011; An et al., 2007). Villani et al. (2007) pointed out thermo-desorption at 250 to 300 °C appeared to be the optimum temperature to avoid size dependent effect. We take 270 °C as the heating temperature, which is just in the middle the optimum range of 250-300 °C recommended by Villani et al. (2007). In fact, previous research using different temperature in the range of 250-300 °C, for examples, 270 °C at Budapest, Hungary (Enroth et al., 2018), 300 °C at Athens, Greece (Mendes et al., 2018), 280 °C at Hyytiälä, Finland (Häkkinen et al., 2012), and 250°C at Nanjing, China (Cui et al., 2021), which we did not express clearly in the original manuscript.

We have reorganized the section "Sampling site and experimental setup" thoroughly. We added more information about this campaign, including sampling site, the configuration of instruments, the aerosol inlet, instrument model, the operation

parameter, and so on. Please see Lines 94-169 in the revised manuscript for detail. The information about sampling was just shown briefly:

The particle number size distribution and mass concentrations of the main chemical composition of non-refractory $PM_1$ were simultaneously measured on the observation platform located on the roof of the CAMS building, where ambient aerosols were pumped through a $PM_{10}$ impactor (URG Corporation at a flow rate of 16.7 L/min) and were then dried to less than 30 % RH using an automatic aerosol dryer. The dried sample was then split through a manifold to different instruments, including the Tandem Scanning Mobility Particle Sizer (TSMPS, TROPOS, Germany) and a High Resolution Time-of-Flight Aerosol Mass Spectrometer (HR-ToF-AMS, Aerodyne Research, Inc., USA). For the VH-TDMA system, a silica dryer and a Nafion dryer in series were used to dry the sample air to less than 30 % RH.

A couple of typos or inaccurate phrases:
- First sentence: replace "aerosols" with ambient aerosol particles
**R:** We have replaced "aerosols" by "ambient aerosol particles" in the revised manuscript as suggested.
- 3 Data analysis, first sentence: add "electrical" before "mobility"
**R:** We have add "electrical" before "mobility" in the revised manuscript as suggested.
- Page 14 row 332: "peaked" should be "peak" right?
**R:** Yes, we changed it as suggested.

**References.**

An, W. J., Pathak, R. K., Lee, B.-H., and Pandis, S. N.: Aerosol volatility measurement using an improved thermodenuder: Application to secondary organic aerosol, J. Aerosol Sci., 38, 305-314, https://doi.org/10.1016/j.jaerosci.2006.12.002, 2007.

Cui, F., Pei, S., Chen, M., Ma, Y., and Pan, Q.: Absorption enhancement of black carbon and the contribution of brown carbon to light absorption in the summer of Nanjing, China, Atmospheric Pollut. Res., 12, 480-487, https://doi.org/10.1016/j.apr.2020.12.008, 2021.

Enroth, J., Mikkilä, J., Németh, Z., Kulmala, M., and Salma, I.: Wintertime hygroscopicity and volatility of ambient urban aerosol particles, Atmos. Chem. Phys., 18, 4533-4548, https://doi.org/10.5194/acp-18-4533-2018, 2018.

Feng, T., Wang, Y., Hu, W., Zhu, M., Song, W., Chen, W., Sang, Y., Fang, Z., Deng, W., Fang, H., Yu, X., Wu, C., Yuan, B., Huang, S., Shao, M., Huang, X., He, L., Lee, Y. R., Huey, L. G., Canonaco, F., Prevot, A. S. H., and Wang, X.: Impact of aging on the sources, volatility, and viscosity of organic aerosols in Chinese outflows, Atmos. Chem. Phys., 23, 611-636, https://doi.org/10.5194/acp-23-611-2023, 2023.

Häkkinen, S. A. K., Äijälä, M., Lehtipalo, K., Junninen, H., Backman, J., Virkkula, A.,

Nieminen, T., Vestenius, M., Hakola, H., Ehn, M., Worsnop, D. R., Kulmala, M., Petäjä, T., and Riipinen, I.: Long-term volatility measurements of submicron atmospheric aerosol in Hyytiälä, Finland, Atmos. Chem. Phys., 12, 10771-10786, https://doi.org/10.5194/acp-12-10771-2012, 2012.

Huffman, J. A., Ziemann, P. J., Jayne, J. T., Worsnop, D. R., and Jimenez, J. L.: Development and characterization of a fast-stepping/scanning thermodenuder for chemically-resolved aerosol volatility measurements, Aerosol Sci. Tech., 42, 395-407, https://doi.org/10.1080/02786820802104981, 2008.

Johnson, G. R., Ristovski, Z. D., D'Anna, B., and Morawska, L.: Hygroscopic behavior of partially volatilized coastal marine aerosols using the volatilization and humidification tandem differential mobility analyzer technique, J. Geophys. Res., 110, https://doi.org/10.1029/2004JD005657, 2005.

Mendes, L., Gini, M. I., Biskos, G., Colbeck, I., and Eleftheriadis, K.: Airborne ultrafine particles in a naturally ventilated metro station: Dominant sources and mixing state determined by particle size distribution and volatility measurements, Environ. Pollut., 239, 82-94, https://doi.org/10.1016/j.envpol.2018.03.067, 2018.

Tritscher, T., Dommen, J., DeCarlo, P. F., Gysel, M., Barmet, P. B., Praplan, A. P., Weingartner, E., Prévôt, A. S. H., Riipinen, I., Donahue, N. M., and Baltensperger, U.: Volatility and hygroscopicity of aging secondary organic aerosol in a smog chamber, Atmos. Chem. Phys., 11, 11477-11496, 10.5194/acp-11-11477-2011, 2011.

Villani, P., Picard, D., Marchand*, N., and Laj, P.: Design and Validation of a 6-Volatility Tandem Differential Mobility Analyzer (VTDMA), Aerosol Sci. Tech., 41, 898-906, https://doi.org/10.1080/02786820701534593, 2007.

---

## Author Comment (AC2)

**Response to Reviewers' comments**

We thank the reviewers for their thoughtful and constructive comments that help us improve the manuscript substantially. We have revised the manuscript accordingly. Listed below is our point-to-point response in blue to each comment that was offered by the reviewers. We hope that our revised manuscript will now be suitable for publication in ACP.

**Response to Reviewer #2**

Yu et al., describes an interesting study on the hygroscopicity and volatility of aerosols in Beijing urban areas during four weeks' field campaign using a Volatility Hygroscopicity Tandem Differential Mobility Analyzer (VH-TDMA) system, a twin scanning mobility particle sizer, a high-resolution time-of-flight aerosol mass spectrometer and other meteorological instruments. The volatile shrink factor (VSF) and the hygroscopic growth factor (HGF) are reported for aerosols having different sizes: 50 nm, 80nm, 110 nm and 150 nm. Furthermore, the authors try to display the size-resolved probability function on the HGF and VSF determined by two Scanning Mobility Particle sizers (SMPS) implicated in the VH-TDMA system. The bimodal distribution has been shown for both two probability functions. During the field campaign, one clean period and three pollution periods are identified. The authors report also the relation between the hygroscopicity and volatility for submicron aerosols. The back trajectories are also carried out for this observation site. According to the bimodal distribution of size-resolved HGF and VSF, the authors suggest that the presence of submicron aerosols in Beijing urban areas is probably under the external mixture due to different sources.

**R:** The authors thank the reviewer's positive comments.

**The manuscript brings a comparatively interesting report but the presentation way, the data analysis methodology and all results related to the multi-charge effect (including HGF-PDFs, VSF-PDFs, kappa and relation between hygroscopicity and volatility) need to be carefully reviewed before the submission. 1ˢᵗ major comment:** the presentation of the measurement campaign is insufficient. A general configuration of all instruments need to be well described, especially how the aerosols are sampled for different instruments. Only a short presentation has been done to indicate the measurement period and the sampling location. All instruments and data should be summarized in this section. However, several fragments could be found in the section of results and discussions, such as meteorological information, back trajectories analysis, etc. Furthermore, the instrumentational information on the three DMAs and two CPCs implicated in the VH-TDMA system is missing, i.e., the model, working conditions... The authors really need to re-organize this section.

**R:** Thanks the reviewer's constructive suggestions. We have reorganized this section thoroughly. We added more information about this campaign, including sampling site, the configuration of instruments, the aerosol inlet, instrument model, the operation parameter, and so on. Please see Lines 94-169 in the revised manuscript for detail.

**2ⁿᵈ major comment**: an important part of the conclusions of this manuscript is based on the bimodal distribution of size-resolved HGF and VSF. However, the phenomenon of multicharged effect on the incoming aerosols by the neutralizer is well known. The authors do not provide the size distribution of selected particles by the DMA1 but only suppose that selected particles are monodispersed. The double charged particles potentially correspond to the larger particles which could contribute to the second mode of HGF and VSF probability functions, i.e., it is not clear whether the second mode is contributed by the different hygroscopic or volatile compounds in the aerosols or is contributed to the double charged particles selected by the DMA (generally these double charged particles are larger than what is commanded in the DMA selection). According to the literature, the double charge effect influences importantly the hygroscopicity analysis of SOA (Bouzidi et al., 2022), combustion aerosols (Petters et al., 2009; Wu et al., 2020), oxygenated aerosols (Petters et al., 2007) at the laboratory and also the measurement during the field campaign (Mochida et al., 2010). Kim et al., 2023 reported a influence of 20% on the kCCN. Several studies suggest how to minimize the double charge effect on the CCN counter (Wang et al., 2015) and HTDMA (Barrett et al., 2012; Duplissy et al., 2009; Oxford et al., 2022). The correction of this multiple charge effect is also provided by Kim et al., 2023 and Petters, 2018. The authors need to take into account this effect in order to better illustrate the conclusions.

**R:** Thanks the reviewer's constructive suggestions. We have added a specific section "3.2 Interference of multi-charged aerosol particles" to discuss the effects of multi-charged aerosol particles in Lines 229-261and the evolution particle number size distributions in Fig.2 in the revised manuscript. We also added the average particle number size distributions (PNSDs) and the calculated number fraction for particles carrying different charges during the clean and pollution period in Fig.S1 (also shown below).

[Figure]

**Figure S1.** (a) The average particle number size distributions (PNSDs) observed during the total sampling period (black point curve), the clean period (blue point curve), and the pollution period (red point curve), respectively. (b) The number fraction for particles carrying different charges (k)

calculated based on charge probability, transfer function of DMA and the PNSD during the clean period and pollution period. For each setting diameter, the left bar corresponds to the clean period, while the right bar corresponds to the pollution period.

The relationship of apparent electrical mobility diameters of multi-charged particles after hygroscopic growth with its dry electrical mobility diameter were shown in Fig. S2. Table S2 gave the summary of electrical mobility diameters and their corresponding physical diameters of doubly and triply charged particles, as well as the physical diameters of doubly and triply charged particles after hydration, the apparent electrical mobility diameter after hydration and hygroscopicity growth factor used.

**Comments in details:**

Line 88, what is the reason that authors chose 270 °C for studying the hygroscopicity of non-volatility particles while this temperature is usually set as 300 °C in the literature?

**R:** The heating temperature of 270 °C for ambient aerosol particles was selected based on the volatility characteristics of relevant inorganic salts, organic compounds, black carbon, soot, and sea salt. At this temperature, atmospheric sulfates, nitrates, and most organic compounds are typically volatile, while soot, sea salt, mineral dust and certain organic polymers remain refractory (Enroth et al., 2018; Johnson et al., 2005). Previous studies have shown that the major inorganic compounds in the atmosphere, such as ammonium nitrate and ammonium chloride, completely volatilize below 150 °C, and ammonium sulfate volatilizes completely between 180 °C and 230 °C (Huffman et al., 2008; Feng et al., 2023). Most organic compounds evaporate at relatively lower temperatures (Tritscher et al., 2011; An et al., 2007). Villani et al. (2007) pointed out thermodesorption at 250 to 300 °C appeared to be the optimum temperature to avoid size dependent effect. We take 270 °C as the heating temperature, which is just in the middle the optimum range of 250-300 °C recommended by Villani et al. (2007).

Line 89, in the section "experimental setup", the authors give a description of each instrument used in this study but no information about how aerosols are sampled among different instruments, that is extremely important to validate the results. For example, were these instruments connected to a common sampling inlet or separately? It is suggested to introduce globally the campaign at the very beginning of this section, including the sampling site, the instruments, sampling conditions and the sources of all meteorological data.

R: Thanks the reviewers' constructive suggestions. We have reorganized this section thoroughly. We added more information about this campaign, including sampling site, the configuration of instruments, the aerosol inlet, instrument model, the operation parameter, and so on. Please see section 2 Sampling site and experimental setup in Lines 94-169 in the revised manuscript.

Line 97, can authors provide a reference which gives more detailed information on this VH-TDMA (TROPOS, Germany)? Otherwise, authors should provide them in this manuscript. For instance, the model of dryer is missing and at what RH aerosols enter in the neutralizer?

**R:** Thanks the reviewer's suggestions. We have thoroughly reorganized the section 2 sampling site and experimental setup. We provided more information about the VH-TDMA system in Lines 111-147 in the revised manuscript, also shown below briefly regarding the exemplary question:

The particle number size distribution and mass concentrations of the main chemical composition of non-refractory $PM_1$ were simultaneously measured on the observation platform located on the roof of the CAMS building, where ambient aerosols were pumped through a $PM_{10}$ impactor (URG Corporation at a flow rate of 16.7 L/min) and were then dried to less than 30 % RH using an automatic aerosol dryer. The dried sample was then split through a manifold to different instruments, including the Tandem Scanning Mobility Particle Sizer (TSMPS, TROPOS, Germany) and a High Resolution Time-of-Flight Aerosol Mass Spectrometer (HR-ToF-AMS, Aerodyne Research, Inc., USA). For the VH-TDMA system, a silica dryer and a Nafion dryer in series were used to dry the sample air to less than 30 % RH.

The VH-TDMA components mainly consist of three medium-Hauke type differential mobility analyzers (DMA1, DMA2, and DMA3, TROPOS, Germany), two CPCs (CPC1 and CPC2; CPC 3772, TSI, USA), an X-ray aerosol neutralizer (Model 3088; TSI Inc., USA), custom-made Nafion dryers, thermal denuders (TD), and humidifiers.

Line 98, what is the model of the DMA1 and in which conditions it works, for example the sheath flow rate? Same questions for the DMA2 and DMA3.
**R:** All three DMAs are medium–Hauke type. All DMAs were operated with a sheath-to-aerosol flow ratio of 5:1. For DMA1, the sheath flow was 10 L min$^{-1}$ with an aerosol flow of 2 L min$^{-1}$, while for DMA2 and DMA3, the sheath flow was 5 L min$^{-1}$ and the aerosol flow was 1 L min$^{-1}$. These information has been added Lines 113, 135-137 in the revised manuscript.

Line 98, what is the model of two CPCs and what size range they measure?
**R:** The model of two CPCs is TSI CPC 3772. The measured size range varied with the selected size from DMA1 and measurement mode of hygroscopicity and volatility. These information has been added in Lines 114, 137-139 in the revised manuscript. The detailed information of measured size range was added to Table S1 in the supplement, also shown below.

**Table S1.** Summary of the size range measured by the DMA after hydration or volatilization for 50-150nm particles under three modes.

| CPC number | Mode | Diameter(nm) | Start of scan(nm) | End of scan(nm) |
|---|---|---|---|---|
| 1 | V-TDMA | 50 | 10 | 120 |
| 1 | V-TDMA | 80 | 10 | 120 |
| 1 | V-TDMA | 110 | 10 | 230 |
| 1 | V-TDMA | 150 | 10 | 230 |
| 2 | H-TDMA | 50 | 30 | 180 |
| 2 | H-TDMA | 80 | 30 | 180 |
| 2 | H-TDMA | 110 | 60 | 360 |
| 2 | H-TDMA | 150 | 60 | 360 |
| 2 | VH-TDMA | 50 | 10 | 120 |
| 2 | VH-TDMA | 80 | 10 | 120 |
| 2 | VH-TDMA | 110 | 10 | 260 |
| 2 | VH-TDMA | 150 | 10 | 260 |

Line 151 and line 156, authors need to declare clearly how the Dp (Troom, RHdry) is defined for the data treatment in this manuscript.

**R:** We revised the two sentences to clarify it in Lines 171-181 in the revised manuscript, also shown below:

The hygroscopic growth factor (HGF) is defined as the ratio of the particle's electrical mobility diameter under a given relative humidity to its diameter under dry condition at room temperature:

$$HGF = \frac{D_p(T_{room}, RH)}{D_p(T_{room}, RH_{dry})}$$

In this study, $D_p$ ($T_{room}$, $RH_{dry}$) is the particle diameter selected by DMA1 at room temperature and an RH below 30 %, while $D_p$ ($T_{room}$, RH) is the diameter of the same particle after being humidified at an RH of 90 %.

The volatile shrink factor (VSF) is defined as the ratio of the particle's electrical mobility diameter under a given temperature to its diameter at room temperature under dry condition:

$$VSF = \frac{D_p(T, RH_{dry})}{D_p(T_{room}, RH_{dry})}$$

In this study, $D_p$ (T, $RH_{dry}$) is the particle diameter at a set heating temperature of 270 °C, which lies in the middle of the optimum temperature range of 250-300 °C to avoid size dependent effect (Villani et al., 2007).

Line 161, for having a size-resolved HGF-PDF, how the Dp (Troom, RHdry) is

defined, the size that the DMA1 selected or a probability density distribution of "monodisperse" aerosols selected by the DMA1? In both case, the multi-charge effect is non-negligible. It is necessaire to provide the size distribution of aerosols selected by the DMA1 or to consider the size distribution of sampled aerosols from site to evaluate this multicharged effect (Duplissy et al., 2009; Kim et al., 2023).

**R:** Thanks the reviewer's constructive suggestions. HGF-PDF was retrieved using the TDMAinv program package (Gysel et al., 2009) in IGOR Pro software, in which the probability density distribution of "monodisperse" aerosols selected by the DMA1 was considered.

Based on the particle number size distribution measured by TSMPS, charging probability and transfer function of DMA (Petters, et al., 2018; Wiedensholer, 1988), we have evaluated the multi-charged effects. We have added a specific section "3.2 Interference of multi-charged aerosol particles" to discuss the effects of multi-charged aerosol particles in Lines 229-261 and the evolution particle number size distributions in Figure 2 in the revised manuscript. Our results indicate that the number fraction of singly charged particles among the 50 nm, 80 nm, 110 nm, and 150 nm particles selected by DMA1 was 94 %, 89 %, 84 %, and 84 %, respectively, during the clean period, whereas during the pollution periods, the value were 91 %, 81 %, 74 %, and 72 %, respectively.

We also added the average particle number size distributions (PNSDs) and the calculated number fraction for particles carrying different charges during the clean and pollution period in Fig. S1. The relationship of apparent electrical mobility diameters of multi-charged particles after hygroscopic growth with its dry electrical mobility diameter was shown in Fig.S2. Table S2 gave the summary of electrical mobility diameters and their corresponding physical diameters of doubly and triply charged particles, as well as the physical diameters of doubly and triply charged particles after hydration, the apparent electrical mobility diameter after hydration and hygroscopicity growth factor used.

Line 165, the same question as the previous. How the Dp (Troom, RHdry) is defined for having a VSF-PDF?

**R:** VSF-PDF was retrieved using the TDMAinv program package (Gysel et al., 2009) in IGOR Pro software, in which the probability density distribution of "monodisperse" aerosols selected by the DMA1 was considered. Dp ($T_{room}$,$RH_{dry}$) represents the particle size selected by DMA1 with RH low than 30% at room temperature.

Line 165, the method how to provide the VHGF-PDF is missing. Since authors have observed the binormal distribution of the non-volatile core of aerosols after heated, how $D$p(270 °C, RHdry) is defined for obtaining your results (for example the VHGF-PDF in figure 11), using the first mode size or the second mode size or the

probability density distribution?

**R:** Thank you for the reviewer's suggestions. We added the information of how to present VHGF-PDF, which was retrieved using the TDMAinv program package (Gysel et al., 2009) in IGOR Pro software just like HGF-PDF, in which the probability density distribution of "monodisperse" aerosols selected by the DMA1 was considered (Lines 200-208 in the revised manuscript).

Line 183, authors really need to take into account the multi-charge effect before calculating the σHGF-PDF.

**R:** Thanks the reviewer's suggestions. We have added a specific section "3.2 Interference of multi-charged aerosol particles" to discuss the effects of multi-charged aerosol particles in Lines 229-261 and the evolution particle number size distributions in Fig. 2 in the revised manuscript. We also added the average particle number size distributions (PNSDs) and the calculated number fraction for particles carrying different charges during the clean and pollution period in Fig. S1. The relationship of apparent electrical mobility diameters of multi-charged particles after hygroscopic growth with its dry electrical mobility diameter was shown in Fig. S2. Table S2 gave the summary of electrical mobility diameters and their corresponding physical diameters of doubly and triply charged particles, as well as the physical diameters of doubly and triply charged particles after hydration, the apparent electrical mobility diameter after hydration and hygroscopicity growth factor used (also shown below). As can be seen from the below table, the apparent mobility diameter of doubly or triply charged particle could be either larger or smaller by a few percent (less than 12 % for size range of 50-150 nm) compared to singly charged particle after hydration. The multi-charged particles could result in the two modes in HGF-PDF or VSD-PDF broadening. Our results indicate that singly charged particles constitute the highest proportion among the particles selected by DMA1 in this study. Specifically, during the clean period, the proportion of singly charged particles among the 50 nm, 80 nm, 110 nm, and 150 nm particles selected by DMA1 were 94%, 89%, 84%, and 84%, respectively. During the pollution period, these proportions were 91%, 81%, 74%, and 72%, respectively. Since singly charged particles dominated in our study, the effect of multiply charged particles should be insignificant, although the number fraction of singly charged particles for some sizes during pollution periods was slightly lower than 80 %.

| | | Electrical mobility diameter (nm) | Physical diameter for different charged particles (nm) | | | HGF used | | | Physical diameter after hydration (nm) | | | Apparent mobility diameter after hydration (nm) | | | Ratio of apparent mobility diameter after hydration (nm) | |
|---|---|---|---|---|---|---|---|---|---|---|---|---|---|---|---|---|
| | | | Singly | Doubly | Triply | Singly | Doubly | Triply | Singly | Doubly | Triply | Singly | Doubly | Triply | Doubly/Singly | Triply/Singly |
| Suppose HGF of the multi-charged particles was the same as that of singly charged particles with the same electrical mobility | The mean HGF of total sampling period are used | 50 | 50 | 72.9 | 91.6 | 1.15 | 1.15 | 1.15 | 57.5 | 83.8 | 105.3 | 57.5 | 57.2 | 57 | 99.5% | 99.1% |
| | | 80 | 80 | 119.2 | 152.4 | 1.24 | 1.24 | 1.24 | 99.2 | 147.8 | 189.0 | 99.2 | 97.8 | 96 | 98.6% | 96.8% |
| | | 110 | 110 | 167.7 | 218.5 | 1.30 | 1.30 | 1.30 | 143.0 | 218.0 | 284.1 | 143 | 139.8 | 137.8 | 97.8% | 96.4% |
| | | 150 | 150 | 235.4 | 313.5 | 1.36 | 1.36 | 1.36 | 204.0 | 320.1 | 426.4 | 204 | 198.5 | 194.5 | 97.3% | 95.3% |
| Suppose HGF of the multi-charged particles was the same as that of particles of the same physical diameter | The mean HGF of total sampling period are used | 50 | 50 | 72.9 | 91.6 | 1.15 | 1.22 | 1.26 | 57.5 | 88.9 | 115.4 | 57.5 | 60.5 | 62 | 105.2% | 107.8% |
| | | 80 | 80 | 119.2 | 152.4 | 1.24 | 1.31 | 1.36 | 99.2 | 156.2 | 207.3 | 99.2 | 103.1 | 105 | 103.9% | 105.8% |
| | | 110 | 110 | 167.7 | 218.5 | 1.30 | 1.36 | 1.36 | 143.0 | 228.1 | 297.2 | 143 | 145.8 | 143.1 | 102.0% | 100.1% |
| | | 150 | 150 | 235.4 | 313.5 | 1.36 | 1.36 | 1.36 | 204.0 | 320.1 | 426.4 | 204 | 198.5 | 194.5 | 97.3% | 95.3% |
| | The mean HGF of clean period are used | 50 | 50 | 72.9 | 91.6 | 1.15 | 1.19 | 1.22 | 57.5 | 86.8 | 111.8 | 57.5 | 59.1 | 60.2 | 102.8% | 104.7% |
| | | 80 | 80 | 119.2 | 152.4 | 1.21 | 1.27 | 1.32 | 96.8 | 151.4 | 201.2 | 96.8 | 100.1 | 102.2 | 103.4% | 105.6% |
| | | 110 | 110 | 167.7 | 218.5 | 1.26 | 1.32 | 1.32 | 138.6 | 221.4 | 288.4 | 138.6 | 141.8 | 139.8 | 102.3% | 100.9% |
| | | 150 | 150 | 235.4 | 313.5 | 1.32 | 1.32 | 1.32 | 198.0 | 310.7 | 413.8 | 198 | 192.8 | 189.8 | 97.4% | 95.9% |
| | The mean HGF of pollution period are used | 50 | 50 | 72.9 | 91.6 | 1.15 | 1.25 | 1.31 | 57.5 | 91.1 | 120.0 | 57.5 | 61.9 | 64.2 | 107.7% | 111.7% |
| | | 80 | 80 | 119.2 | 152.4 | 1.28 | 1.39 | 1.44 | 102.4 | 165.7 | 219.5 | 102.4 | 108.8 | 110.3 | 106.3% | 107.7% |
| | | 110 | 110 | 167.7 | 218.5 | 1.36 | 1.44 | 1.44 | 149.6 | 241.5 | 314.6 | 149.6 | 153.6 | 150.3 | 102.7% | 100.5% |
| | | 150 | 150 | 235.4 | 313.5 | 1.44 | 1.44 | 1.44 | 216.0 | 339.0 | 451.4 | 216 | 208.3 | 204.1 | 96.4% | 94.5% |

Line 196, could authors provide the size distribution of aerosols measured by the PNSD to show the seven NPF events?

**R:** Thanks the reviewer's suggestions. The evolution of particle number size distribution during the measurement period has been added in Fig. 2 in the revised manuscript. The mean particle number size distribution during the NPF days and the particle number size distribution during each NPF day have been added in Fig. S6 in the supplement, also shown below:

[Figure]

Line 209, authors need to take into account the multi-charge effect before giving the conclusion.

**R:** Thanks the reviewer's suggestions. Based on the particle number size distribution measured by TSMPS, charging probability and transfer function of DMA, we have evaluated the multi-charged effects. We have added a specific section "3.2

Interference of multi-charged aerosol particles" to discuss the effects of multi-charged aerosol particles in Lines 229-261 in the revised manuscript. We also added the average particle number size distributions (PNSDs) and the calculated number fraction for particles carrying different charges during the clean and pollution period in Fig. S1. Our results indicate that the number fraction of singly charged particles among the 50 nm, 80 nm, 110 nm, and 150 nm particles selected by DMA1 was 94 %, 89 %, 84 %, and 84 %, respectively, during the clean period, whereas during the pollution period, the value were 91 %, 81 %, 74 %, and 72 %, respectively. Since singly charged particles dominated in our study, the effect of multiply charged particles should be insignificant, although the number fraction of singly charged particles for some sizes during pollution periods was slightly lower than 80 %.

The relationship of apparent electrical mobility diameters of multi-charged particles after hygroscopic growth with its dry electrical mobility diameter was shown in Fig. S2. Table S2 gave the summary of electrical mobility diameters and their corresponding physical diameters of doubly and triply charged particles, as well as the physical diameters of doubly and triply charged particles after hydration, the apparent electrical mobility diameter after hydration and hygroscopicity growth factor used. As can be seen from Table S2, the apparent mobility diameter of doubly or triply charged particle could be either larger or smaller by a few percent (less than 12 % for size range of 50-150 nm) compared to singly charged particle after hydration. That is to say, the multi-charged particles result in the two modes in HGF-PDF or VSD-PDF broadening, which is already reflected by the retrieved PDF if any multi-charged particle exists. Therefore, we think the conclusion that HGF-PDFs and VSF-PDFs exhibit clear bimodal distributions for different particles is still valid.

Line 216, in the case of non-consideration of multi-charge effect, the variation (number and mode size) of ambient aerosol size distribution due to clean and/or pollution events could contribute to the variations in the HGF-PDFs and VSF-PDFs. It is suggested that authors provide not only the mean particle size distribution during the clean period as Figure S4, but also that as function as time during the whole measurement campaign.

**R:** Thanks the reviewer's suggestions. We have added the evolution of particle number size distribution during the whole measurement campaign in Fig.2b in the revised manuscript, also shown below:

[Figure]

Line 240, "narrower" is not a quantitatively description. Could authors provide the geometrical standard deviation of two modes for different cases in order to illustrate this conclusion?

**R:** Thanks the reviewers' suggestions. We calculated the standard deviation of the MH modes for different particle sizes, they were $0.074 \pm 0.025$, $0.088 \pm 0.023$, $0.09 \pm 0.022$, and $0.094 \pm 0.024$ for particle sizes of 50 nm, 80 nm, 100 nm and 150 nm, respectively. Although the full width at half maxima of 150 nm is smaller than that of other sizes, we delete this sentence in the revised manuscript.

Line 265, figure 3 shows the mean HGF-PDF and the mean VSF-PDF of particles having different sizes. In figure 3 a, how to explain the HGF of left part in the first mode (around ¼ visibly) is lower than 100%?

**R:** Thanks the reviewer's suggestions. The DMA classifies aerosol particles based on their mobility in an electric field. Only particles in a narrow mobility range can pass through the classifier exit slit for later analysis. In other words, the DMA selects particles approximately in a triangular distribution centered around the mobility centroid $Z^{*}_{p}$ (corresponding to set diameter). The range that can pass through the DMA is proportional to the products of mobility centroid $Z^{*}_{p}$ and the ratio of sheath-to-sample flow ratio. Therefore, some particles with diameter smaller than the set diameter in a narrow range could be measured later. If these particles exhibit low or negligible hygroscopicity, will result in only a slight increase or no change in particle size after hygroscopicity, which will result in the HGF is below 1. This phenomenon has also been observed in previous studies (Cai et al., 2018; Wang et al., 2018; Enroth et al., 2018, Mikhailov et al., 2004; Shingler et al., 2016).

Line 268, the general presentation of the table 1 is missing. In the table 1, authors need to well define the "Total". Does "Total" means the whole period from 11 October to 6 November 2023? If yes, it is interesting that the mean k of 50 nm in "Total" is slightly higher than that in both "Clean" and "Pollution". Is it significant large in the period which is not "Clean" and "Pollution"?

**R:** We added the general description of Table 1 in Lines 315-317 in the revised

manuscript. We are sorry that we did not express the meaning of "Total", "Clean" and "Pollution" clearly in the original manuscript. "Total" means the whole period from 11 October to 6 November 2023. Clean and pollution period is about 38%, and 31% of the total measurement time, respectively. Just like what you expected, the $\kappa_{mean}$ for 50 nm particles is large in the period which is not "Clean" or "Pollution" period. For example, the $\kappa_{mean}$ for 50 nm particles for the whole day of Oct. 17 is $0.073 \pm 0.030$ which drive the average value up.

Line 314, "AMS results show that nitrate is the main inorganic component of PM1 (Fig. 8a), further supporting this viewpoint." It is known that nitrate signals in the AMS could represent inorganic nitrate compounds or organic nitrate compounds. Graeffe et al., 2023 reported how to estimate the organic nitrate. Authors should provide more information such as NO+/NO2+ ratio to support this conclusion.

**R:** We appreciate the reviewer's valuable suggestion. We are sorry that we did not offer the information of organic nitrate from the measurement at our site. Fortunately, the study by Xu et al., (2018) has focused on the chemical composition and pollution processes of $PM_1$ during autumn in Beijing based on HR-ToF-AMS, and also reported the organic nitrate mass concentration and the contribution to total nitrate. The result showed that the organic nitrate mass ranged from 0.01 to 6.8 μg/m³, with an average of $1.0 \pm 1.1$ μg/m³, and organic nitrate components accounted for 10% of the total nitrate mass. Therefore, inorganic nitrate is dominated in the total nitrate mass. We have added this information in Lines 397-399 in the revised manuscript.

Line 329, "During the clean period, particle volatility increased dramatically around 10:00 LT (VSF mean decreased) along with the occurrence of NPF events, indicating that the newly formed matter was more volatile." Replace "around" by "starting from".

**R:** Changed as the reviewer suggested.

Line 348, "Notably, external mixing was more apparent during the night and early morning, especially for 150 nm particles during the clean period and 50 nm particles during the pollution period (Fig. 5d and 5e). This phenomenon could be attributed to the reduced boundary layer height, which leads to the accumulation of nonvolatile particulate matter emissions (e.g. BC, soot aggregates) from cooking or vehicles emissions." Authors need to take into account the multi-charge effect which could contribute to the VSF-PDF the same way as the external mixing. It is suggested to combine off-line technics such as Transmission electron microscopy to have a direct prove on the presence of soot particles and/or sea salts mentioned later.

**R:** Thanks the reviewer's suggestions. Based on the PNSD, charge probability and transfer function of DMA, we calculated the number fraction of different charge

particles in Fig. S1. As can be seen from this figure, the number fraction of singly charged particle for electrical mobility of 50 nm during the pollution period is 91%, while 85% for 150nm during the clean period. Duplissy et al. (2009) mentioned that the singly charged aerosol particles dominate (>80%) and the HTDMA data analysis is straight forward and not biased by the multiply charged aerosol particles. So, we think the multi-charge effect contribute less to this external mixing. Just like the reviewer's suggestion, transmission electron microscopy is a direct way to observe soot particles and/or sea salts in the non-volatile particles after heating. Unfortunately, we did not collect off-line samples in this study, we will consider TEM in future research.

Line 467, it is suggested to present ZSR relation and the method of calculating HGFcoating in the section of "data analysis".
**R:** Thanks the reviewer's suggestions. We have moved the ZSR relation and the method of calculating $HGF_{coating}$ to "Data analysis" section in Lines 193-199 in the revised manuscript.

Line 479, "As shown in Fig. 8c, variations in HGFcoating basically similar to those hygroscopicity of unheated particles (HGFmean) (Fig.S7) and exhibit significant size dependency" What is the difference between figure.S7 and figure.2 and figure.S2 on the part of HGF? What is the interest to show the same data by using size independent time series (figure.S7) and using size-resolved probability function time series (figure.2 and figure.S2)? What is the procedure authors calculate HGFcoating, by using HGFmean or HGF-PDF or some other method for the core and for the original aerosols?
**R:** Thanks the reviewer's suggestions. Fig. 2 and Fig.S2 (Change the original Fig. S2 to Fig. S4) showed the hygroscopic growth factor probability density function (HGF-PDF) for different selected sizes, which give us more information on the evolution of different modes, and provide insights into the mixing state of particles. However, Fig.S7 (Change the original Fig. S7 to Fig. S8) just showed their mean hygroscopic growth factor ($HGF_{mean}$) with time, which directly compared the $HGF_{mean}$ value of different sizes.

$HGF_{coating}$ was calculated using formula 6 in the revised manuscript (also shown below) according to the Zdanovskii - Stokes - Robinson (ZSR) relation. This relation supposes the mean hygroscopic growth factor of a particle ($HGF_{mean}$) can be estimated from the HGFs of its components according to their volume fractions. In this study, only non-volatile core and volatile coating were considered.

$$HGF_{mean}^3 = VFR * HGF_{core}^3 + (1 - VFR) * HGF_{coating}^3$$

Where HGFmean was the mean hygroscopicity calculated from the HGF-PDF for

certain size particles, while HGFcore was the ratio of mean hygroscopicity from VHGF-PDF to that from VSF-PDF for the same selected size particles, VFR is the volume fraction of non-volatile core in the selected particles, is the the cubic of VSF. This information could be found in Lines 186-199 in the revised manuscript.

Line 501, according to the figure 11, the distribution of core particles (after the heating of 270 °C) is not monodisperse. How authors calculate HGF-core using equation 3?

**R:** Thanks the reviewer's suggestions. VSF-PDF and VHGF-PDF were retrieved using the TDMAinv program package (Gysel et al., 2009) in IGOR Pro software, in which the probability density distribution of "monodisperse" aerosols selected by the DMA1 was considered. VSF-PDF in Fig.11 showed a minor non-volatile (NV) mode and a major very volatile (VV) mode (marked by $D_p$(270 °C, $RH_{dry}$)), whereas there are three modes marked by $D_{p1}$(270 °C, 90 %RH), $D_{p2}$(270 °C, 90 %RH), $D_{p3}$(270 °C, 90 %RH) in VHGF-PDF. Since the NV mode was smaller than the $D_{p3}$(270 °C, 90 %RH) mode, thus we assume the three modes in VHGF-PDF results from the hygroscopic growth of VV mode of the particles after heated. Taking Event I as an example, the particle size at the peak of the VV mode in the VSF-PDF was 75 nm ($D_p$ (270 °C, $RH_{dry}$)). After hygroscopic growth, the three modes peak at $D_{p1}$(270 °C, 90 %RH)=77 nm, $D_{p2}$(270 °C, 90 %RH)=96 nm, $D_{p3}$(270 °C, 90 %RH)=135 nm. Thus, the hygroscopic growth factor of the hygroscopic components in the non-volatile core was then calculated as follows: $D_{p1}/D_p$ = 77/75 = 1.03, $D_{p2}/D_p$ = 96/75 = 1.28, and $D_{p3}/D_p$ = 135/75 = 1.80. It is no doubt this estimate, although with some uncertainty, gives some valuable information about the complexity of aerosol hygroscopicity and volatilization.

Line 522, it is the first time to see "VHGF-PDF". The explanation is required.

**R:** Thanks the reviewer's suggestions. Volatility hygroscopic growth factor (VHGF) is defined as the ratio of a particle's electrical mobility diameter after humidification following heating to its diameter under dry condition. This has been added to the "Parameters derived from VH-TDMA " section in Lines 182-185 in the revised manuscript, also shown below

Volatility hygroscopic growth factor (VHGF) is defined as the ratio of a particle's electrical mobility diameter after humidification following heating to its diameter under dry condition:

$$VHGF = \frac{D_p(T, RH)}{D_p(T_{room}, RH_{dry})}$$

In this study, $D_p$(T, RH) is the diameter of particle after humidification at 90% RH following heating at 270 °C.

Line 535, the time scale with multi colors is not explained.

**R:** Thanks the reviewer's suggestions. The color scale represents the time of air mass backward trajectory arrived at the sampling site. The back trajectories were calculated every two hours. We have added this information in the caption of Fig.10 in Lines 627-628 in the revised manuscript.

**References.**

An, W. J., Pathak, R. K., Lee, B.-H., and Pandis, S. N.: Aerosol volatility measurement using an improved thermodenuder: Application to secondary organic aerosol, J. Aerosol Sci., 38, 305-314, https://doi.org/10.1016/j.jaerosci.2006.12.002, 2007.

Cai, M., Tan, H., Chan, C. K., Qin, Y., Xu, H., Li, F., Schurman, M. I., Liu, L., and Zhao, J.: The size-resolved cloud condensation nuclei (CCN) activity and its prediction based on aerosol hygroscopicity and composition in the Pearl Delta River (PRD) region during wintertime 2014, Atmos. Chem. Phys.,18, 16419-16437, https://doi.org/10.5194/acp-18-16419-2018, 2018.

Cui, F., Pei, S., Chen, M., Ma, Y., and Pan, Q.: Absorption enhancement of black carbon and the contribution of brown carbon to light absorption in the summer of Nanjing, China, Atmospheric Pollut. Res., 12, 480-487, https://doi.org/10.1016/j.apr.2020.12.008, 2021.

Duplissy, J., Gysel, M., Sjogren, S., Meyer, N., Good, N., Kammermann, L., Michaud, V., Weigel, R., Martins dos Santos, S., and Gruening, C.: Intercomparison study of six HTDMAs: results and recommendations, Atmos. Meas. Tech., 2, 363-378, https://doi.org/10.5194/amt-2-363-2009, 2009.

Enroth, J., Mikkilä, J., Németh, Z., Kulmala, M., and Salma, I.: Wintertime hygroscopicity and volatility of ambient urban aerosol particles, Atmos. Chem. Phys., 18, 4533-4548, https://doi.org/10.5194/acp-18-4533-2018, 2018.

Feng, T., Wang, Y., Hu, W., Zhu, M., Song, W., Chen, W., Sang, Y., Fang, Z., Deng, W., Fang, H., Yu, X., Wu, C., Yuan, B., Huang, S., Shao, M., Huang, X., He, L., Lee, Y. R., Huey, L. G., Canonaco, F., Prevot, A. S. H., and Wang, X.: Impact of aging on the sources, volatility, and viscosity of organic aerosols in Chinese outflows, Atmos. Chem. Phys., 23, 611-636, https://doi.org/10.5194/acp-23-611-2023, 2023.

Gysel, M., Mcfiggans, G. B., and Coe, H.: Inversion of tandem differential mobility analyser (TDMA) measurements, J. Aerosol Sci., 40, 134-151, https://doi.org/10.1016/j.jaerosci.2008.07.013, 2009.

Häkkinen, S. A. K., Äijälä, M., Lehtipalo, K., Junninen, H., Backman, J., Virkkula, A., Nieminen, T., Vestenius, M., Hakola, H., Ehn, M., Worsnop, D. R., Kulmala, M., Petäjä, T., and Riipinen, I.: Long-term volatility measurements of submicron atmospheric aerosol in Hyytiälä, Finland, Atmos. Chem. Phys., 12, 10771-10786, https://doi.org/10.5194/acp-12-10771-2012, 2012.

Huffman, J. A., Ziemann, P. J., Jayne, J. T., Worsnop, D. R., and Jimenez, J. L.: Development and characterization of a fast-stepping/scanning thermodenuder for chemically-resolved aerosol volatility measurements, Aerosol Sci. Tech., 42, 395-407, https://doi.org/10.1080/02786820802104981, 2008.

Johnson, G. R., Ristovski, Z. D., D'Anna, B., and Morawska, L.: Hygroscopic behavior of partially volatilized coastal marine aerosols using the volatilization and humidification tandem differential mobility analyzer technique, J. Geophys. Res., 110, https://doi.org/10.1029/2004JD005657, 2005.

Mendes, L., Gini, M. I., Biskos, G., Colbeck, I., and Eleftheriadis, K.: Airborne ultrafine particles in a naturally ventilated metro station: Dominant sources and mixing state determined by particle size distribution and volatility measurements, Environ. Pollut., 239, 82-94, https://doi.org/10.1016/j.envpol.2018.03.067, 2018.

Mikhailov, E., Vlasenko, S., Niessner, R., and Pöschl, U.: Interaction of aerosol particles composed of protein and saltswith water vapor: hygroscopic growth and microstructural rearrangement, Atmos. Chem. Phys., 4, 323-350, https://doi.org/10.5194/acp-4-323-2004, 2004.

Petters, M. D.: A language to simplify computation of differential mobility analyzer response functions, Aerosol Sci. Tech., 52, 1437-1451, https://doi.org/10.1080/02786826.2018.1530724, 2018.

Shingler, T., Sorooshian, A., Ortega, A., Crosbie, E., Wonaschütz, A., Perring, A. E., Beyersdorf, A., Ziemba, L., Jimenez, J. L., and Campuzano-Jost, P.: Ambient observations of hygroscopic growth factor and f (RH) below 1: Case studies from surface and airborne measurements, J. Geophys. Res., 121, 13661-13677, https://doi.org/10.1002/2016JD025471, 2016.

Tritscher, T., Dommen, J., DeCarlo, P. F., Gysel, M., Barmet, P. B., Praplan, A. P., Weingartner, E., Prévôt, A. S. H., Riipinen, I., Donahue, N. M., and Baltensperger, U.: Volatility and hygroscopicity of aging secondary organic aerosol in a smog chamber, Atmos. Chem. Phys., 11, 11477-11496, 10.5194/acp-11-11477-2011, 2011.

Villani, P., Picard, D., Marchand*, N., and Laj, P.: Design and Validation of a 6-Volatility Tandem Differential Mobility Analyzer (VTDMA), Aerosol Sci. Tech., 41, 898-906, https://doi.org/10.1080/02786820701534593, 2007.

Wang, X., Shen, X. J., Sun, J. Y., Zhang, X. Y., Wang, Y. Q., Zhang, Y. M., Wang, P., Xia, C., Qi, X. F., and Zhong, J. T.: Size-resolved hygroscopic behavior of atmospheric aerosols during heavy aerosol pollution episodes in Beijing in December 2016, Atmos. Environ., 194, 188-197, https://doi.org/10.1016/j.atmosenv.2018.09.041, 2018.

Wiedensohler, A. and Fissan, H. J.: Aerosol charging in high purity gases, J. Aerosol. Sci., 19, 867-870, https://doi.org/10.1016/0021-8502(88)90054-7, 1988.

Xu, P., Zhang, J., Ji, D., Liu, Z., Tang, G., Jiang, C., and Wang, Y.: Characterization of submicron particles during autumn in Beijing, China, J. Environ. Sci., 63, 16-27,

https://doi.org/10.1016/j.jes.2017.03.036, 2018.

---

## Author Response (AR2)

**Response to Reviewers' comments**

We thank the editor and reviewers for their thoughtful and constructive comments that help us improve the manuscript substantially. We have revised the manuscript accordingly. Listed below is our point-to-point response in blue to each comment that was offered by the reviewers. We hope that our revised manuscript will now be suitable for publication in ACP.

**Response to Editor:**

The authors revised their MS in accordance to the reviewers' comments. The MS is virtually ready for publication. I just wonder if the authors would go through the figures containing several colour shades considering the comment from the file validation process.

R: Thanks for the suggestion. We have carefully reviewed the figures with multiple color shades. Using the Coblis – Color Blindness Simulator, we have adjusted the color schemes of Fig. S5 and 8. The revised figures were added to the revised manuscript and supplement.

Please ensure that the colour schemes used in your maps and charts allow readers with colour vision deficiencies to correctly interpret your findings. Please check your figures using the Coblis – Color Blindness Simulator (https://www.color-blindness.com/coblis-color-blindness-simulator/) and revise the colour schemes accordingly with the next file upload request. -> Fig. S5, 8

R: Thanks for the suggestion. We have adjusted the color schemes of Fig. S5 and 8 and verified them using the Coblis – Color Blindness Simulator to ensure accurate interpretation by readers. The revised figures have been included in the updated manuscript and supplement and are also shown below.

[Figure]

[Figure]